# Future Therapeutic Strategies for Alzheimer’s Disease: Focus on Behavioral and Psychological Symptoms

**DOI:** 10.3390/ijms252111338

**Published:** 2024-10-22

**Authors:** Kyoung Ja Kwon, Hahn Young Kim, Seol-Heui Han, Chan Young Shin

**Affiliations:** 1Department of Pharmacology, School of Medicine, Konkuk University, Seoul 05029, Republic of Korea; neuro11@kku.ac.kr; 2Center for Neuroscience Research, Institute of Biomedical Science and Technology, Konkuk University, 120 Neungdong-ro, Gwangjin-gu, Seoul 05029, Republic of Korea; hykimmd@gmail.com (H.Y.K.); alzdoc@naver.com (S.-H.H.); 3Department of Neurology, Konkuk Hospital Medical Center, 120-1 Neungdong-ro, Gwangjin-Gu, Seoul 05030, Republic of Korea

**Keywords:** Alzheimer’s disease, neuropsychiatric symptoms, psychomotor symptoms, social function, pharmacotherapy

## Abstract

Alzheimer’s disease (AD) is a progressive, degenerative brain disorder that impairs memory and thinking skills, leading to significant economic and humanistic burdens. It is associated with various neuropsychiatric symptoms (NPS) such as anxiety, agitation, depression, aggression, apathy, and psychosis. NPSs are common in patients with AD, affecting up to 97% of individuals diagnosed with AD. The severity of NPS is linked to disease progression and cognitive decline. NPS in Alzheimer’s disease leads to increased morbidity, mortality, caregiver burden, earlier nursing home placement, and higher healthcare costs. Despite their significant impact, clinical research on NPS in AD is limited. In clinical settings, accurately distinguishing and diagnosing NPS related to AD remains a challenge. Additionally, conventional treatments for NPS in AD are often ineffective, highlighting the need for new therapies that target these specific symptoms. Understanding these comorbidities can aid in early diagnosis and better management of AD. In this review, we provide a summary of the various neurological and psychiatric symptoms (NPS) associated with AD and new candidates under development for the treatment of NPS based on their therapeutic targets and mechanisms. On top of the conventional NPS studied so far, this review adds recent advancements in the understanding of social functional impairment in AD. This review also provides information that can contribute to the advancement of studies and translational research in this field by emphasizing therapeutic targets and mechanisms of action focused on AD-related NPS rather than conventional mechanisms targeted in AD drug development. Above all, considering the relative lack of research in this new field despite the importance of clinical, medical, and translational research, it may increase interest in NPS in AD, its pathophysiological mechanisms, and potential therapeutic candidates such as molecules with antioxidant potential.

## 1. Introduction

Alzheimer’s disease (AD) is a progressive neurological disorder that affects cognitive function, primarily memory and thinking skills. It is the most common cause of dementia, accounting for 60–80% of cases.

AD imposes a substantial economic burden on society, encompassing direct (such as medical and non-medical expenses) and indirect costs (related to informal caregiving, lost productivity, and intangible costs). The total societal burden of AD exceeds USD 958 billion worldwide and is projected to increase significantly by 2050 [1]. Beyond its economic impact, AD also imposes a huge humanistic burden on individuals affected by it, their caregivers, and society. This burden encompasses emotional, psychological, and social challenges such as reduced quality of life, loss of independence, changes in relationships, caregiver stress and burnout, stigma, and overall decline in well-being.

AD has predominantly been researched with a focus on cognitive function and memory decline in terms of diagnosis, management, and treatment development. However, over the decades, patients, families, and healthcare providers have recognized that various behavioral, neurological, psychological, and psychiatric symptoms can accompany AD, both before and after its onset. These symptoms include depression, apathy, anxiety, agitation, aggression, personality changes, sleep disturbances, fatigue, psychosis (hallucinations, delusions, and paranoia), sensory impairment, and motor dysfunction. Collectively, these symptoms, which will be defined as neuropsychiatric symptoms (NPS) in this article, are observed in approximately one-third of individuals with dementia residing in the community and up to 80% of those in long-term care facilities supervised by nursing staff [2].

Neuropsychiatric symptoms (NPS) are fundamental features of Alzheimer’s disease and related dementias, with nearly all individuals diagnosed with AD developing NPS during the disease progression. The prevalence of NPS in AD can vary depending on factors such as disease stage, age of onset, and genetic predispositions. While NPS typically becomes more pronounced in the later stage of AD, population-based studies have shown that these symptoms often manifest in the very early stages, including the prodromal stage, such as MCI [3,4,5,6,7]. Among NPS, depression and apathy are the most frequently observed in patients with MCI and early AD, often accompanied by verbal and physical agitation. As AD advances, symptoms such as delusions, hallucinations, and aggression become more common, while apathy remains the most persistent and prevalent NPS across all stages of the disease. In addition, disruptions in circadian sleep–wake cycles become more pronounced in patients with AD compared to normal aging processes. Notably, patients with MCI who experience depression are at a significantly higher risk of progressing to AD. Conversely, individuals with mild behavioral impairment are more likely to develop dementia, even if their cognitive function is within normal range. Symptoms like agitation, apathy, anxiety, disinhibition, euphoria, and irritability may have stronger associations with the onset of MCI than depression. Therefore, further research is needed to investigate the difference in prevalence associated with these factors.

Analysis of comorbidities in patients with AD has revealed a combination of neurological, psychiatric, and peripheral disorders [3,8,9] with significant involvement of reactive oxygen species and the anti-oxidative system, inflammation, obesity, and hypertension as determining factors [9,10], and the trajectory of comorbidity could be different from that of the aging population without AD symptoms [11]. A study based on a Spanish cohort over 6 years period reported a wide range of comorbidities in patients with AD. Approximately 55.1% of patients had 9–12 comorbidities, followed by 32.8% of patients with 5–8 comorbidities [12]. The most common comorbidities include hypertension, neurotic disorders, personality disorders, and other nonpsychotic disorders, as well as metabolic disorders and immune disorders [12].

The occurrence of various comorbidities, especially neuropsychiatric comorbidities, in AD varies depending on disease onset, stage, and pathological state. Sometimes, the presence of a neuropsychiatric condition obscures the detection of AD and vice versa. Therefore, identifying these neurological comorbidities and understanding their characteristics can aid in the early diagnosis and treatment of AD. Additionally, the occurrence of these co-symptoms can be influenced by various factors, such as age at disease onset, level of education, and gender. Utilizing this information can help to develop personalized treatment approaches [3,8,11,13]. Furthermore, neurological comorbidities associated with AD are often not well controlled by conventional antidepressants and antipsychotics that are typically used for patients with individual disorders [3]. This highlights the urgent need for the development of new therapies that can specifically modulate NPS, indicating the presence of disease-specific mechanisms underlying NPS in AD [14]. The new treatment approach may encompass several unconventional and innovative approaches, such as genetic considerations related to AD manifestation and pathophysiology, nonpharmacologic approaches such as psychosocial paradigms, and exploration of alternative therapeutic targets such as 5-HT2A receptor antagonists/inverse agonists and 5-HT6 receptor antagonists for AD-related psychosis [14].

In this review, we aimed to provide a brief summary of the various neurological and psychiatric symptoms associated with AD, followed by a description of candidate compounds for clinical trials aimed at modulating these symptoms and their mechanisms of action. Specifically, we seek to offer information that can contribute to the advancement of research and translational studies in this field by highlighting therapeutic targets and mechanisms focusing on improving AD-related neurological and psychiatric symptoms rather than conventional mechanisms targeted in drug development. Additional attention should also be paid to social interaction deficits in AD and its pathophysiological mechanism, along with potential therapeutic candidates considering the relative scarcity of research in this emerging field, albeit of clinical, medical, and translational importance (Figure 1).

## 2. NPS in AD

### 2.1. Depression

Depression in AD is characterized by persistent feelings of worthlessness or excessive/inappropriate guilt, often accompanied by a pervasive loss of interest or pleasure in nearly all activities (American Psychiatric Association, 2013, [18]). This condition significantly diminishes the quality of life, impairs activities of daily living, exacerbates cognitive decline, heightens the likelihood of physical aggression, and may hasten nursing home admission and increase mortality rates among patients with AD [19]. These symptoms serve as prognosticators of worse outcomes in patients with AD, including diminished quality of life, increased risk of self-harm and suicidality, and accelerated cognitive decline. Depressive symptoms are highly prevalent in AD, with a major depression prevalence of 14.8% [20]. Over 40% of individuals diagnosed with AD experience significant depressive symptoms over the course of their disease, a markedly higher proportion compared to the general population, which ranges from 10 to 15% [21,22]. Conversely, patients with depression have an elevated risk of developing AD [23]. While the precise reasons for the heightened incidence of depression in patients with AD have not been fully elucidated, it is clear that the physical and psychological distress associated with AD progression significantly contributes to the onset of depression.

In sporadic AD, tau protein abnormalities often affect key brain regions early on, including the dorsal raphe nucleus (serotonin) and locus coeruleus (norepinephrine) [24]. The loss of these diffuse systems can manifest as changes in behavior, particularly depression, anxiety, and agitation. There are indications of alterations in the serotonergic system contributing to depression in patients with AD, such as single nucleotide polymorphisms (SNPs) in 5-HT2A and 2C receptors as well as in the serotonin transporter (SERT), along with changes in the expression levels of 5-HT1A, 2A, 2C, and SERT. However, results have been inconsistent and may vary across different brain regions and patients.

Currently, selective serotonin reuptake inhibitors (SSRIs) are the primary pharmacological treatment for depression in patients with AD owing to their relatively low side-effect profiles. However, large-scale studies and meta-analyses of carefully conducted clinical trials have revealed no significant therapeutic effects of SSRIs compared to placebo [25,26]. This lack of efficacy may be attributed to the challenges in applying common psychiatric diagnostic criteria and clinical features to a population with neurodegenerative diseases [27]. These findings underscore the pressing need to investigate the precise pathophysiological mechanisms of depression in AD and to develop more targeted medications.

### 2.2. Anxiety

Anxiety in AD is characterized by excessive worry, fear, restlessness, and irritability. These symptoms can exacerbate cognitive decline, hinder social interactions, and increase caregiver burden. Anxiety is distressing and common among individuals with AD, with a meta-analysis conducted in 2021 revealing a pooled prevalence of up to 39% [28]. Similar to depression, anxiety has been linked to a heightened risk of dementia and cognitive impairment [29,30].

Because of the potential for adverse effects in the elderly population, benzodiazepines are not recommended as the preferred treatment method. Currently, selective serotonin reuptake inhibitors (SSRIs), serotonin–norepinephrine reuptake inhibitors (SNRIs) such as venlafaxine, and serotonergic atypical anxiolytics such as buspirone are commonly prescribed to treat anxiety in patients with AD. Recent meta-analytical studies have indicated significant therapeutic effects of certain drugs, such as citalopram, compared to control groups [31]. However, concerns persist regarding the relative efficacy and overall advantage of these drugs, considering their common side effects, particularly gastrointestinal and neurological side effects, which are more prevalent in elderly patients.

### 2.3. Apathy

Apathy in AD is characterized by a decline in motivation and interest across emotional, goal-directed behavior, and cognitive activity domains. Managing and supporting patients with apathy can significantly increase caregiver burden and service utilization, as individuals may struggle to make decisions or pursue life-sustaining goals. Apathy is the most common NPS in AD, with a prevalence exceeding 49%, is often linked with functional impairment, and has become a focal point for treatment interventions [32,33]. Importantly, apathy can persist throughout the disease progression [34,35]. However, distinguishing between apathy and depression poses a challenge. While apathy and depression may coexist or be mistaken for one another, antidepressants have not demonstrated efficacy in treating apathy and may even exacerbate it. Regrettably, evidence from treatment studies is generally of low quality, largely because of the variability in the measurement tools employed. Secondary analyses have shown the potential effectiveness of cholinesterase inhibitors but not of memantine, although prospective studies remain limited. Based on a recent Cochrane review, the most compelling evidence thus far supports the use of methylphenidate at a daily dose of 20 mg, making it the recommended first-line agent in cases where cholinesterase inhibitors are not used [36]. Various other interventions, such as Ginkgo biloba, modafinil, and SSRIs, have been investigated in the treatment of apathy in AD, but their efficacy varies, and some may be associated with side effects or offer limited benefits [19]. Apathy represents a critical aspect of NPS in AD, highlighting the urgent need for novel treatment targets and therapeutic advancements in this area.

### 2.4. Agitation and Aggression

Agitation and aggression are prevalent in AD, and their frequency escalates as the disease advances, thereby augmenting the economic burden and challenges in caregiving. Aggression occurs in approximately 30% of patients with AD and may increase to 60% among patients with AD living in long-term care facilities [19,37,38]. Agitation, a nonspecific symptom, leads to variability in the study population. While definitions may vary across studies, manifestations of agitation and aggression in patients with AD include restlessness, excessive fidgeting, verbal abuse, shouting, and even physical violence. Recently, the International Psychogeriatric Association (IPA) introduced a syndromic definition, categorizing agitation into excessive motor activity, verbal aggression, and physical aggression [39]. Antipsychotics and benzodiazepines have been used to address these symptoms, albeit with heightened risks of adverse reactions, such as falls, cognitive decline, and delirium, particularly among the elderly. Furthermore, discontinuing antipsychotic treatment abruptly can exacerbate agitation and aggression, thereby establishing an unfavorable cycle [40].

Likewise, the utilization of mood stabilizers such as valproic acid is linked to diverse adverse effects and varying degrees of effectiveness [41]. Brexpiprazole, an atypical antipsychotic, was approved by the FDA specifically for treating agitation symptoms in AD based on comparative clinical trials that demonstrated its superiority over other antipsychotics in alleviating these symptoms [42]. Considering the pharmacological mechanism of brexpiprazole, which entails partial agonism of serotonin 5-HT1A and dopamine D2 receptors and antagonism of serotonin 5-HT2A receptors, further investigation is warranted to ascertain whether its specific pharmacology confers superior therapeutic potential in this particular patient cohort.

### 2.5. Psychosis

Psychosis, characterized by delusions, hallucinations, and disorganized thought and speech, typically worsens as AD progresses and is often accompanied by agitation and aggression in affected patients. Psychosis in AD leads to confusion, fear, distress, paranoia, mistrust, and behavioral disturbances, impacting communication, caregiving, relationships, and patient management. Moreover, psychosis in AD is associated with poor clinical outcomes, including accelerated cognitive decline. It is estimated that up to 50% of patients with AD will experience some form of psychosis during the disease course, usually in later stages, although its severity and duration vary among individuals [43]. Despite being relatively less investigated, patients with AD with aphasia and psychosis may exhibit further challenges due to disorganized thought and speech, complicating social functioning as the disease advances [44].

When evaluating evidence for the treatment of agitation and psychosis in dementia, it is crucial to recognize that different assessment measures may vary in capturing symptoms and response to treatment. Similar to agitation, the IPA and the International Society to Advance Alzheimer’s Research and Treatment NPS Professional Interest Area are in the process of developing a revised syndromic definition for psychosis in dementia [45].

There have been mixed outcomes with both traditional and newer versions of antipsychotics such as aripiprazole, risperidone, quetiapine, brexpiprazole, and anticonvulsants. Similar to other medications used for NPS, the administration of antipsychotics should be meticulously determined and monitored because of the wide range of potential adverse reactions associated with this class of medication, including, but not limited to, increased sedation, cognitive decline, extrapyramidal side effects, and cardiovascular risks.

Numerous new agents are currently under investigation for managing agitation and psychosis in patients with AD. Dextromethorphan–quinidine has demonstrated efficacy in treating pseudobulbar affect and is currently being explored for its potential in AD treatment [46]. Dextromethorphan acts as a sigma-1 and mu-opioid receptor agonist, inhibits serotonin and norepinephrine transporters, and also exhibits antagonist properties at NMDA and nicotinic α3β4 receptors. When combined with quinidine, it extends its half-life, which would otherwise be too short. Additionally, pimavanserin, which is already approved for treating psychosis in individuals with Parkinson’s disease, is also undergoing investigation for its application in AD [47].

### 2.6. Sleep Disturbances

Most patients with AD typically encounter various sleep difficulties, including insomnia, excessive daytime sleepiness, nighttime awakenings, and alterations in sleep–wake patterns. Research suggests that approximately 45% of patients with AD experience some form of sleep disturbance [48,49]. Recent meta-analyses have highlighted that insomnia is associated with an increased risk of cardiovascular and mental disorders, including AD [50].

While changes in sleep architecture and physiology are common in normal aging, pronounced sleep issues in AD not only disrupt the patients’ and caregivers’ quality of life but also significantly impact the disease’s pathophysiology and progression. Sleep disruption, including sleep apnea, is recognized as a prominent factor contributing to the onset of AD [51,52]. Studies indicate that sleep deprivation in animals mirrors the behaviors observed in patients with AD [53] and can elevate amyloid beta levels in the cerebrospinal fluid after just one day of sleep disturbance in middle-aged men [54]. Reduced sleep may lead to prolonged neural activity, resulting in increased amyloid beta production, reduced clearance, and subsequent plaque formation, thereby perpetuating a cycle of sleep deficits [51].

Polysomnographic studies reveal that patients with AD exhibit various sleep abnormalities compared to healthy individuals, including decreased total sleep time, reduced sleep efficiency, alterations in slow–wave sleep (SWS) and rapid eye movement (REM) sleep, as well as increased sleep latency, wake time after sleep onset, number of awakenings, and REM latency [49]. Although evidence supporting alterations in sleep EEG frequency components and sleep spindles in AD is limited, these changes have also been noted [49].

Given reports suggesting the potential impact of sleep management on AD pathology and functional performance, understanding the characteristics of sleep deprivation in patients with AD and its underlying mechanisms is vital for developing more effective and safe treatment methods [55,56]. Common pharmacological treatments for sleep disturbances in AD include melatonin and its receptor activators (e.g., ramelteon), antidepressants (such as trazodone), and orexin antagonists (like suvorexant and lemborexant) [55]. However, clinical trials assessing the efficacy of these medications on sleep disturbances in patients with AD have shown varying results, with some indicating only modest improvements and significant individual differences in response. In many of these clinical trials, it remains uncertain whether there’s improvement in certain aspects of sleep architecture, such as the frequency of nighttime awakenings, compared to more obvious parameters like total sleep duration and the onset time to fall asleep. Non-pharmacological interventions, including behavioral strategies, environmental adjustments, and cognitive-behavioral therapy for insomnia, are often preferred by researchers and physicians due to their lower risk of side effects and drug interactions [57]. This highlights the need for the development of more effective therapeutic approaches to address this unmet medical need.

### 2.7. Social Processing

Social interaction plays a crucial role in the pathophysiology of AD. Patients with AD often exhibit reduced social engagement [58,59]. Disease-related phenotypes, such as language problems, personality changes, and irritability, may contribute to this reduced social processing as well as the neurobiological changes hampering the proper functioning of brain circuits responsible for regulating social interaction [60]. Active and prolonged social participation may reduce the risk of developing dementia by 30–50%, although the exact causality remains unclear. It is postulated that social engagement enhances cognitive reserve and promotes brain maintenance, potentially through stress reduction and improved cerebrovascular health [61]. While interventions targeting social participation have demonstrated improvements in cognition, long-term studies specifically addressing dementia risk reduction are rare, with short-term comparative studies not consistently providing positive results [61].

The link between social behaviors and cognitive deficits in animal models of AD has been extensively investigated. Social isolation exacerbates cognitive impairment [62], although the precise neurobiological mechanisms remain elusive [62,63]. Possible mechanisms include increased production of Aβ peptide, tau protein phosphorylation [64,65], elevated oxidative stress, inflammatory reactions [66], impaired anti-inflammatory responses [67], altered synaptic plasticity (reduction in brain-derived neurotrophic factors (BDNF) [68,69], and myelination changes [70]. Ali et al. investigated the role of social isolation as a risk factor for AD development in animal models by isolating rats for 4 weeks [62]. Prolonged social isolation led to neurological damage in the brain, evidenced by significant increases in markers such as Aβ, AChE, MDA, TNF-α, and IL-1β, along with decreases in SOD, TAC, BDNF, and monoamines. These findings have been corroborated by histopathological changes observed in various brain regions, which are more pronounced in AD models. Isolation also enhanced DNA fragmentation induced by AD, providing insights into the impact of social relationships on cognitive health [62].

In line with the results described above, N-acetyl cysteine administration has been reported to decrease cognitive impairment in socially isolated mice [71]. Increased social interaction has also been associated with the normalization of memory and cognitive function deficits in an AD animal model, achieved through elevated expression of BDNF. Interestingly, BDNF, which plays a crucial role in synaptic plasticity and regulation of cognitive function, can be epigenetically influenced by social interactions [65,72]. Early life-time social enrichment, such as the communal nest paradigm in mice, leads to higher NGF and BDNF levels in the brain. Adult mice exposed to this enrichment exhibit greater social propensity and achieve a more prompt social dominance behavioral profile, which demonstrates how early social enrichment can influence social behavior and neurotrophic factor levels in the brain, highlighting the importance of social interactions in normal brain functioning [73].

Given these mechanisms, enhancing social interaction may contribute to better disease outcomes in human patients with AD. However, further research is needed to fully investigate this possibility.

Among the six domains of neurological function outlined by the Research Domain Criteria (RDoC) of the NIH, only the aspect concerning the “multi-dimensional modulation of social processing” remains resistant to clinicopharmacological modulation. As a result, cognitive-behavioral modulation and social support programs emerge as the sole viable options for enhancing social interaction in patients with AD at the moment. The development of drugs aimed at improving sociability, similar to social enhancement therapeutics used for conditions like autism spectrum disorder, may hold promise for modulating social impairments in patients with AD, which could be the crucial step to determine whether enhancing sociability positively impacts the course of AD.

### 2.8. Implication from the NPS in AD

There are several implications arising from the aforementioned studies examining the relationship between NPS and AD, as well as AD-related dementia. Firstly, the comorbidity of NPS in AD is influenced by co-existing variables such as disease onset time, disease stage, gender, socio-economic status, family history, and so forth. Secondly, the presence of NPS not only affects the care and management of patients with AD but also influences the pathophysiology and prognosis of AD. Thirdly, AD neuropathology correlates with the severity of NPS in general, with varying degrees of correlation among different symptoms. Certain NPS are more common in specific subsets of AD-related disorders (ADRD). For example, hallucinations were more frequent in individuals with both AD and Lewy body disease (LBD) neuropathology compared to LBD alone. Apathy and disinhibition were common in individuals with behavioral variants of frontotemporal lobar degeneration and hippocampal sclerosis.

Lastly, commonly used medications, while generally effective, do not consistently provide sufficient therapeutic efficacy or safety profiles for patients with AD. This phenomenon cannot be solely explained by the age profile of patients with AD, indicating that sophisticated mechanisms are at play in modulating specific neurobehavioral, neurobiological, and neurochemical determinants underlying the regulation of specific NPS. For example, Consentino et al. suggested that social cognition in patients with AD is a separable construct from general cognition [74]. It is evident that neuronal death, synaptic degeneration, and functional deficits may work in concert with neurochemical and neurobiological deficits secondary to changes occurring in specific regions of the AD brain. Carefully designed and conducted clinical trials sometimes yield promising therapeutic effects against certain NPS in the AD population, suggesting that it might be possible to target specific neurobiological mechanisms relevant to NPS in patients with AD (Table 1).

## 3. Treatment Pipelines, Targets, and Mechanisms of Action

### 3.1. An Antagonist of NMDA Receptor

#### 3.1.1. Dextromethorphan

Recently, dextromethorphan has gained attention as a candidate substance for treating AD and NPS accompanied by AD (Table 2). Previously used as a cough suppressant, its protective mechanisms are known to reduce brain cell damage and suppress inflammation. These properties have drawn the interest of researchers as they might help inhibit neuronal degeneration and delay the progression of the disease in patients with AD. Chen et al. reported that individuals who used dextromethorphan had a lower risk of developing dementia compared to those who did not use it, even after adjusting the age and other variables [75]. Notably, dextromethorphan not only reduces the onset of AD in proportion to its amount of usage but also significantly decreases dementia onset caused by various risk factors for AD, including liver and kidney diseases, diabetes, and cardiovascular diseases [75]. This suggests broad potential for dextromethorphan in mitigating dementia risk across different health conditions.

In addition to its neuroprotective effects, dextromethorphan possesses pharmacological profiles as an NMDA receptor antagonist, σ-1 receptor agonist, serotonin and norepinephrine transporter inhibitor, and nicotinic σ3β4 receptor antagonist, all of which have been implicated in the modulation of neurological disorders, although the efficacy is relatively weak (Figure 2). For example, modulation of the σ-1 receptor has been implicated in regulating symptoms and disease progression in various neurodegenerative and neurodevelopmental disorders. While the precise mechanisms are not yet clearly demonstrated, they appear to be pleiotropic. These mechanisms include calcium shuttling into mitochondria, increased ATP production, trafficking of NMDA receptors, modulation of synaptic transmission through SK channels, and regulation of channels such as L-type Ca^2+^ channels, voltage-gated Na^+^ channels, and A-type K^+^ channels, etc [76]. By acting as a σ-1 receptor agonist, dextromethorphan inhibits presynaptic glutamate release, potentially contributing to the modulation of synaptic transmission and providing a neuroprotective mechanism [77]. In addition to neurochemical and neurophysiological changes, dextromethorphan also modulates glial survival and inflammatory activation. It reduces microglia-mediated dopaminergic neuron degeneration, inhibits NOX-2 transcription, and decreases the production of TNF-α, NO, and superoxide anion. Madeira et al. suggest that dextromethorphan reduces lesion size and neuronal cell death, possibly by lowering microglial activation. Furthermore, it promotes oligodendrocyte and progenitor cell proliferation and differentiation, affecting myelination processes [15].

**Table 2 ijms-25-11338-t002:** Candidates and mechanism of action under development for the treatment of AD-related neuropsychiatric symptoms.

Candidate Drugs	Target	Mechanism of Action	Target NPS	Disease Stage of AD	Clinical Status	References
Lumateperone (Caplyta)	Neurotransmitter receptors	A 5-HT2A antagonist, a SERT inhibitor, a DRD2 antagonist, a GluN2B modulator	Depression			[78,79,80,81]
Psilocybin	Neurotransmitter receptors	A modulator of 5-HT1A, 5-HT2A, and 5-HT2C receptors		Prodormal/prodromal-mild	Phase 1	[82,83,84]
Cannabidiol	Neurotransmitter receptors	Cannabinoid, an agonist against 5-HT1A, A2A, and TRP-V1 receptors, anti-inflammatory action	Anxiety	Mild–moderate dementia, prodormal/prodromal-mild	Phase 1 Phase 2	[85,86]
Dextromethorphan	Neurotransmitter receptors	NMDA receptor antagonist; σ-1 receptor agonist, a SERT and NET inhibitor, a nicotinic σ3β4 receptor antagonist	Agitation			[15,75,76,77,87,88]
AVP-923 (Neudexta)	Neurotransmitter receptors	dextromethorphan/ quinidine		Mild–moderate dementia, severe dementia		[87,89,90]
AXS-05	Neurotransmitter receptors	dextromethorphan/ bupropion NMDA receptor antagonist; σ-1 receptor agonist. A SERT and NET inhibitor		Mild–moderate dementia, severe dementia	Phase 3	[91,92,93]
Nabilone	Cannabinoid receptor	A partial agonist against CB1 and CB2 receptor		Mild–moderate dementia, severe dementia	Phase 3	[94,95]
Brexpiprazole	Neurotransmitter receptors	A partial agonist against D2, D3 receptor, and 5-HT1A. A serotonin and dopamine modulator		Mild–moderate dementia, severe dementia		[42,96,97]
AVP-786	Neurotransmitter receptors	NMDA receptor antagonist; σ-1 receptor agonist. A SERT and NET inhibitor		Mild–moderate dementia, severe dementia	Phase 3	[16,98]
Dexmedetomidine	Neurotransmitter receptors	An α2 adrenergic agonist		Mild–moderate dementia, severe dementia	Phase 3	[99,100,101]
JZP541	Cannabinoid receptor	An agonist against CB1 and CB2 receptor		Mild–moderate dementia Severe dementia	Phase 2	[16]
Dronabinol	Cannabinoid receptor	A weak partial agonist against CB1 and CB2 receptor		Mild–moderate dementia	Phase 2	[102,103,104]
IGC-AD1	Cannabinoid receptor	Cannabinoid, a partial agonist against CB1 receptor		Mild–moderate dementia, severe dementia	Phase 2	[16,86]
Prazosin	Neurotransmitter receptors	An α1 adrenergic antagonist			Phase 2b	[105,106]
SCI-110	Neurotransmitter receptors	Tetrahydrocannabinol and palmitoylethanolamide		Mild–moderate dementia	Phase 2	[107,108]
THC-Free CBD	Cannabinoid receptor	An agonist against CB1 and CB2 receptor		Mild–moderate dementia, severe dementia	Phase 2	[109,110]
Masupirdine	Neurotransmitter receptors	A 5-HT6 receptor antagonist		Mild–moderate dementia	Phase 3	[111,112,113,114]
MK-8189	Phosphodiesterase	A PDE10a inhibitor		Mild–moderate dementia	phase 1	[115]
Pimvanserin	Neurotransmitter receptors	A selective inverse agonist of the serotonin 5-HT_2A_ receptor	Psychosis		Phase 3	[47,116,117]
ACP-204	Neurotransmitter receptors	A potent and selective antagonist/inverse agonist of 5-HT2A receptor		Mild–moderate dementia, severe dementia	Phase 2 Phase 3	[15,16,83,118]
KarXT	Cholinergic modulator	A dual M1/M4 muscarinic acetylcholine receptor agonist		Mild–moderate dementia	Phase 3	[119,120,121,122,123]
Seltorexant	Orexin system	A selective antagonist of the orexin-2 receptor	Sleep disturbances	Mild–moderate dementia	Phase 2	[124,125]
SLV	Neurotransmitter receptors	A selective 5-HT6 receptor antagonist	Social processing			[126]
N-acetyl cysteine	Redox system	An antioxidant and glutathione inducer				[71]
BDNF	Neurotrophic factor	A member of the neurotrophin family, TrkB activation				[65,72,73]
Oxytocin	Endocrine system	a nonapeptide hormone, oxytocin receptor activation				[127,128,129]

One drawback of dextromethorphan as a CNS drug candidate is its poor brain penetration and the side effects of both the drug itself and its metabolite [130]. Dextromethorphan is commonly abused due to its euphoric, hallucinogenic, and dissociative properties, especially at high concentrations [131]. Additionally, dextromethorphan may lead to various adverse events, including hypertension, seizures, tachycardia, psychosis, and rhabdomyolysis, depending on the amount ingested.

Due to the strong metabolism of dextromethorphan by the cytochrome P450 (CYP) liver enzyme CYP2D6, it exhibits high first-pass metabolism and low bioavailability, along with substantial individual variation in pharmacokinetic parameters. To enhance its effectiveness, a pharmacological approach combines dextromethorphan with another drug to prevent its metabolism, thereby increasing its bioavailability in the brain. Co-administration of a small dose of quinidine with dextromethorphan decreases the latter’s metabolism, resulting in a 25-fold increase in the concentration of free dextromethorphan. This allows free dextromethorphan to effectively reach the brain and exert its neurological and psychiatric effects [87,88].

Several key clinical trials have evaluated the efficacy of dextromethorphan, often in combination with other compounds like quinidine and bupropion to enhance its efficacy, for treating NPS in AD. The combination with bupropion (AXS-05) demonstrated therapeutic efficacy for depression, while the combination with quinidine (AVP-786, AVP-923) aimed to assess its efficacy for agitation [132,133].

#### 3.1.2. AVP-923

AVP-923 is a combination of dextromethorphan and quinidine, used to increase the bioavailability of dextromethorphan by inhibiting its metabolism via the cytochrome P450 enzyme (CYP2D6) (Table 2). Quinidine allows dextromethorphan to achieve therapeutic levels in the brain, where it can modulate various neurotransmitter systems. An incrementally modified drug combination of dextromethorphan and quinidine has demonstrated significant efficacy in reducing agitation and aggression of patients with AD compared to placebo [39,134]. The drug was generally well-tolerated, although side effects such as falls, diarrhea, and urinary tract infections were reported. The combination of dextromethorphan and quinidine sulfate, administered at a dose of 20 mg/10 mg twice daily (Neudexta or AVP-923), is approved for treating pseudobulbar affect in the United States and the European Union. Evidence suggesting a potential effect of dextromethorphan/quinidine for agitation in dementia comes from controlled clinical trial data in non-demented patients with pseudobulbar affect [87]. However, subsequent studies yielded mixed results.

#### 3.1.3. AVP-786

Following successful findings in the phase 2 trial with AVP-923, the FDA granted a fast-track designation to the deuterated sister compound, AVP-786, for direct investigation in phase 3 trials targeting agitation in AD [133] (Table 2). Deuteration of dextromethorphan was shown to reduce the amount of quinidine needed to achieve an effective plasma concentration of free dextromethorphan, thereby minimizing drug-drug interaction and cardiac side effects typically associated with quinidine. Apart from the pharmacokinetic alteration, deuteration of dextromethorphan did not show to affect the selectivity and affinity for brain receptors implicated in neuropsychiatric effects. The two completed phase 3 trials with AVP-786, sponsored by Avanir Pharmaceuticals (TRIAD-1 (NCT02442765) and TRIAD-2 (NCT02442778), yielded contradictory findings, potentially due to differences in clinical study design—one study adopted a sequential parallel comparison design, while the other followed a more conventional study approach—highlighting the need for large-scale confirmatory clinical trials [98].

#### 3.1.4. AXS-05

Another combination, dextromethorphan/bupropion, is FDA-approved for treating major depression in adults. This combination demonstrated promising results in a Phase 2 trial for agitation in dementia and also in a couple of Phase 3 trials [91] (Table 2). Bupropion is a norepinephrine and dopamine reuptake inhibitor commonly used as an antidepressant, which serves to increase the bioavailability of dextromethorphan by acting as a strong CYP2D6 inhibitor, similar to quinidine. It was reported that add-on treatment of dextromethorphan with bupropion XL leads to rapid-acting antidepressant effects within 48 h in treatment-resistant depression [92,93,135]. In the multicenter, double-blind, placebo-controlled phase 2/3 study (ADVANCE-1) using dextromethorphan/bupropion (AXS-05) in 366 patients with AD and agitation, the dextromethorphan/bupropion combination showed better efficacy compared to bupropion alone and placebo. AXS-05 was well-tolerated, and the combination showed promise for its ability to manage NPS with fewer side effects compared to traditional antipsychotics. This combination has received FDA designation for breakthrough therapy. More recent results from the AXS-05 phase 3 trial (ACCORD study, ClinicalTrials.gov Identifier: NCT04797715), which composed of an open-label period followed by a randomized placebo-controlled study period in patients with a diagnosis of probable AD and clinically meaningful agitation associated with the disease, demonstrated significant delay and reduction of AD-related agitation relapse. Preliminary results suggest that AXS-05 may reduce the likelihood of agitation relapse, indicating its potential for long-term management of NPS in AD.

### 3.2. An Inverse Agonist and Antagonist 5-HT2A Receptor

#### 3.2.1. Pimavanserin

Another actively investigated candidate for NPS in AD is pimavanserin, an atypical antipsychotic (Table 2). Pimavanserin acts as an inverse agonist and antagonist at serotonin 5-HT_2A_ receptors, with lesser potency at 5-HT_2C_ receptors and almost negligible affinity against dopamine receptors (Figure 2). It has been approved for treating psychosis associated with PD, sold under the brand name Nuplazid, and is actively being studied for a similar indication in patients with AD [47,116,117]. However, a phase 3 randomized controlled trial (RCT) examining the effect and safety of 20 mg and 34 mg of pimavanserin treatment on the relapse of hallucinations and delusions associated with dementia-related psychosis (NCT03325556) revealed only marginal efficacy of pimavanserin over placebo for this indication. Nonetheless, it demonstrated an excellent safety profile in another phase 3b trial [136], which necessitates further carefully designed studies.

#### 3.2.2. ACP-204

ACP-204 is an inverse agonist and potent antagonist at the 5-HT2A receptor (Figure 2), developed by Acadia Pharmaceuticals for the treatment of hallucinations and delusions associated with AD psychosis (ADP) [16] (Table 2). Currently, ACP-204 is under clinical development in both phase 2 and phase 3 for ADP (NCT06159673, NCT06194799).

### 3.3. A Partial Agonist of Dopamine D2 Receptor and 5-HT1A Receptor

Brexpiprazole (Rexulti), developed by Otsuka and Lundbeck, grained FDA approval in July 2015. It’s an atypical antipsychotic medication used to treat major depressive disorder, schizophrenia, and agitation associated with dementia due to AD [42,96] (Table 2). Brexpiprazole targets noradrenergic, serotonergic, and dopaminergic neurotransmitter systems. It is a partial agonist of the dopamine D2/D3 receptor and 5-HT1A receptor, which is known as a serotonin–dopamine activity modulator (SDAM) [97]. Notably, Brexpiprazole is the first treatment approved by the FDA for agitation associated with dementia due to AD [15].

### 3.4. An Antagonist of 5-HT2A and Dopamine (D1, D2, and D4) Receptor

Lumateperone (ITI-007, ITI-722) received its initial approval from the FDA in the USA in December 2019 for treating schizophrenia in adults. Its therapeutic efficacy is under investigation for bipolar depression, behavioral disorders linked to AD and dementia, sleep disorders, and major depressive disorders [78,79] (Table 2). Pharmacodynamically, lumateperone exhibits potent antagonistic activity at serotonin 5-HT2A receptors and is a serotonin transport (SERT) inhibitor (Figure 2). Additionally, it functions as a presynaptic partial agonist and a postsynaptic antagonist at dopamine D2 receptors, along with being a dopamine D1 receptor-dependent indirect modulator of GluN2B receptors [80]. Given this multi-target activity, lumateperone has the potential to modulate depressive symptoms and irritability by attenuating overt neural activity, although direct clinical evidence is needed in the future.

### 3.5. Glutamate Receptor Modulator and an Inositol Monophosphatase Inhibitor

Another candidate for modulation of NPS in AD is lithium, which is clinically used for bipolar disorders and manic psychosis. Although the exact mechanism of action of lithium as a mood stabilizer is unknown, it has been shown to modulate glutamate receptors (especially GluR3) and inhibit inositol monophosphatase [137]. In a four-site randomized clinical trial, low-dose lithium failed to significantly alleviate agitation in patients with AD but was associated with global clinical improvement and was generally considered safe [138].

### 3.6. Norepinephrine Modulators

It has been theorized that norepinephrine is functionally overactive compared to other prominent neuromodulators such as dopamine and serotonin in AD, which suggests novel combinations of pharmacological agents to counteract AD [105]. Phase IIb clinical trials using the α1 adrenergic antagonist prazosin to assess efficacy against agitation in patients with AD have been completed, albeit with reduced participant sizes due to the COVID pandemic, and await final report. Similarly, maintaining an adequate level of norepinephrine is proposed to be pivotal in maintaining proper neural function. Therefore, the use of norepinephrine-modulating drugs is suggested as an interesting alternative therapeutic option for AD [106]. Targeting the noradrenergic system in preclinical models using norepinephrine reuptake inhibitors, monoamine oxidase inhibitors, α2 adrenergic antagonists, L-DOPS, tyrosine hydroxylase potentiation, and direct locus coeruleus stimulation has been shown to modulate AD pathophysiology and memory impairment. These mechanisms include reduced inflammation, diminished amyloid burden, neuroprotection, glial modulation, and regulation of neurotrophic factors [106].

### 3.7. Cannabinoid Receptors

Medical cannabis containing CBD and THC, such as Epidiolex and Sativex, has been approved and used for several neurological conditions, such as epilepsy associated with Lennox Gastaut syndrome, Dravet syndrome, Tuberous sclerosis as well as multiple sclerosis [139]. Additionally, various formulations and synthetic varieties of these compounds are actively being investigated for their potential effects on NPS in AD (Table 2).

#### 3.7.1. Nabilone

Nabilone, a synthetic cannabinoid, is used as an antiemetic in patients undergoing cancer treatment. It acts as a partial agonist at CB1 and CB2 receptors, similar to tetrahydrocannabinol (THC) (Figure 2). Nabilone produces dose-dependent mood elevation and psychomotor inhibition comparable to 10 or 20 mg of oral dronabinol (THC). Although nabilone has been implicated in alleviating the symptoms of fibromyalgia, neuropathic pain, and multiple sclerosis, there are reports of postoperative pain aggravation. In a 14 week clinical trial with a crossover design (including a 1 week washout period between nabilone and placebo), nabilone improved agitation, overall behavior, and caregiver distress compared to placebo. While sedation was more pronounced in the nabilone treatment group, it did not significantly limit treatment efficacy [94,95]. Ongoing research includes a phase 3 interventional study with a larger participant group (112) and a phase 2 study investigating agitation in frontotemporal dementia (NCT05742698). Although nabilone is argued to effectively reduce agitation in patients with AD, it may impair cognitive functions in healthy adults.

#### 3.7.2. Dronabinol

Dronabinol, the (-)-trans-enantiomer of THC, improves anorexia induced by HIV and alleviates chemotherapy-induced nausea and vomiting (Figure 2). As early as 1997, dronabinol was suggested to have a beneficial effect on anorexia and disturbed behavior in patients with AD [102], and ongoing clinical trials explore its impact on agitation in dementia and AD [103,104].

#### 3.7.3. JZP541

JZP541 is a botanical cannabinoid drug product containing cannabidiol (CBD), cannabichrome (CBC), and delta-9-tetrahydrocannabinol (THC) (Figure 2). JZP541, developed by Jazz Pharmaceuticals, has undergone a phase 2 clinical trial to evaluate its efficacy and safety as a treatment for irritability associated with Autism Spectrum Disorder (ASD). Recently, JZP541 has been undergoing phase 2 trials to evaluate the safety and therapeutic efficacy for managing agitation in patients with AD (CALM-IT, NCT06014424).

#### 3.7.4. IGC-AD1

IGC-AD1 is a natural THC-based formulation administered in microdoses. It combines a CB1 receptor partial agonist with anti-neuroinflammatory properties and an inflammasome inhibitor targeting the upregulation of inflammasome-3 (Figure 2). Although not yet published, it has been argued that IGC-AD1 demonstrated safety and tolerability in a phase 1 clinical trial. Interim phase 2 results suggest a clinically and statistically significant reduction in agitation among patients with AD compared to placebo (NCT05543681). These findings appear more favorable than those for brexpiprazole, but further verification is needed in final, larger-scale trials. The combination of the inflammasome inhibitor is proposed to reduce amyloid-beta plaque buildup and promote mitochondrial function, potentially restoring spatial memory based on in vitro and in vivo preclinical studies.

#### 3.7.5. SCI-110

SCI-110, previously known as THX-110, is another candidate for addressing NPS in AD. SCI-110 is a combination of THC and palmitoylethanolamide (PEA), administered separately as pills. A completed phase 2 open-label trial (NCT05239390) examined the safety, tolerability, and efficacy trend of SCI-110 in patients with AD and agitation. However, a larger study is needed in the future. Recently, the sponsor of SCX-110 initiated clinical trials of SCI-110 for treating Tourette syndrome. Palmitoylethanolamide (PEA) is an endogenous fatty acid amide and lipid modulator. It exerts pleiotropic effects on PPAR-alpha and other nuclear receptors, as well as cannabinoid-like G-coupled receptors GPR55 and GPR119. Although PEA does not directly bind to CB1 and CB2 receptors, it is argued to enhance the activity of anandamide [107,108]. PEA is metabolized by cellular enzymes, including fatty acid amide hydrolase (FAAH) and N-acylethanolamine acid amide hydrolase (NAAA), which play a role in the degradation of endocannabinoids.

#### 3.7.6. Cannabidiol (CBD)

While cannabidiol (CBD) does not directly bind to classical CB1 and CB2 receptors, its efficacy against AD and related NPS is actively under investigation. In an animal model of AD, CBD treatment enhanced cognitive function and protected against Aβ42-induced neurotoxicity [85]. These effects may be related to CBD’s anti-inflammatory action and its engagement with multiple targets, including 5-HT1A, A2A, and TRP-V1 receptors [140]. However, the detailed mechanism should be verified in future studies. An ongoing open-label subchronic clinical trial is evaluating sublingual CBD solutions for treating clinically significant anxiety and agitation in individuals with mild cognitive impairment or mild to moderate AD (NCT04075435). This trial may provide further evidence regarding CBD’s effectiveness in AD and NPS associated with AD.

### 3.8. Cholinergic Modulators

KarXT is a novel dual-action agent that combines Xanomeline and Trospium (Table 2). It selectively targets the M1/M4 muscarinic acetylcholine receptors in the CNS as a partial agonist (Figure 2). Although ligand binding analysis suggests a nearly identical binding affinity for all five different muscarinic receptors, KarXT has shown promising results in clinical trials targeting schizophrenia [119]. Notably, it even addresses cognitive impairment observed in these patients [120,121]. Ongoing phase 3 clinical trials are evaluating KarXT’s efficacy against NPS in AD (ADEPT-1, NCT05511363). KarXT’s primary therapeutic principle is xanomeline, combined with the peripherally restricted muscarinic receptor antagonist trospium chloride. This combination aims to mitigate xanomeline-related adverse events associated with peripheral muscarinic receptors. Given xanomeline’s muscarinic action, it is anticipated that KarXT will mitigate cognitive impairment in AD. Moreover, the therapeutic efficacy of KarXT not only makes it feasible for targeting schizophrenia with cholinergic modulators but also for intervening in NPS associated with AD [122,123]. Recently, KarXT (Cobenfy) gained FDA approval for schizophrenia as the first antipsychotic with a cholinergic mechanism of action.

### 3.9. 5-HT_6_ Receptor Antagonists

The 5-HT_6_ receptor has recently gained attention as a potential therapeutic option in AD. However, in phase 2 clinical trials evaluating the efficacy and safety of the 5-HT_6_ receptor antagonist masupirdine as an adjunct treatment for patients with moderate AD (concomitantly treated with donepezil and memantine), no significance was observed compared to the placebo group [111] (Table 2). The 5-HT_6_ receptor is selectively expressed in the CNS, specifically on the GABAergic interneuron, contributing to the tonic inhibition of neurotransmitter release. Selective blockade of the 5-HT_6_ receptor may improve learning and memory, possibly by increasing acetylcholine and glutamate output [141]. Consequently, 5-HT_6_ receptor antagonists hold promise as novel therapeutic candidates for addressing memory deficits in AD [113,114]. BASED on subgroup analysis showing improvement in the agitation/aggression domain of the Neuropsychiatric Inventory scores during the phase 2 study, masupirdine is now undergoing phase 3 clinical trials for treating NPS in AD [112].

### 3.10. Orexin-2 Receptor Antagonists

Seltorexant is a selective antagonist of the human orexin-2 receptor (Figure 2). The neuropeptide orexin regulates wakefulness, and inhibiting orexin receptor signaling promotes sleep (Table 2). The orexin system also modulates aggressive behaviors, motor behavior, reactivity to stress, anxiety, reward processing, and addictive behaviors. Seltorexant has completed phase 2 clinical trials in probable patients with AD and with clinically significant agitation and aggression (NCT05307692). Additionally, seltorexant is in advanced clinical development for treating depression, where its effectiveness is thought to be related to improving sleep. Given that sleep deprivation plays a critical role in the pathogenesis of AD, there is a mechanistic rationale for using an orexin antagonist to improve symptoms associated with AD. Seltorexant is being investigated as a potential treatment for agitation and aggression in people with AD [124,125].

### 3.11. An Agonist of Alpha-2 Adrenergic Receptor

Dexmedetomidine is an agonist of α2-adrenergic receptors having sedative, anxiolytic, hypnotic, analgesic, and sympatholytic properties, inhibiting the release of norepinephrine (Figure 2). It has anti-inflammatory effects and cognitive-enhancing effects [99,100] (Table 2). It was approved by the FDA for the treatment of agitation in schizophrenia and bipolar disorder in 2022 [101]. Recently, BioXcel Therapeutics conducted a phase 3 clinical trial to evaluate the safety and efficacy of BXCL501, an oral formulation of dexmedetomidine, as a treatment for agitation associated with dementia, including Alzheimer’s disease (TRANQUILITY II, NCT05271552).

### 3.12. PDE10 Inhibitors

Among various subtypes of phosphodiesterase enzymes that degrade cAMP and cGMP, phosphodiesterase type 10 (PDE10) is expressed in the brain, particularly within the striatum, medium spiny neurons, nucleus accumbens, and olfactory tubercle (Table 2). PDE10 emerges as a potential candidate against psychiatric and neurological disorders. Reduced activation of dopamine D1 receptor signaling may contribute to negative symptoms in schizophrenia, and PDE10A, which modulates both dopamine D2- and D1-dependent signaling, is a clinical candidate aimed at improving cognitive and negative symptoms associated with schizophrenia. Additionally, PDE10 is being investigated as a promising therapeutic strategy for psychiatric and neurodegenerative diseases, based on its efficacy in animal models of schizophrenia, Parkinson’s, Huntington’s, and AD [142,143]. While negative results regarding the effect of a PDE10 inhibitor on reward-based effortful behavior and reward learning indices have been published [144], another PDE10a inhibitor, MK-8189, is currently undergoing phase 1 clinical trials in participants with AD with agitation-aggression and/or psychosis [115].

### 3.13. Psychedelic Compounds

Psilocybin, a naturally occurring psychedelic compound found in certain mushroom species, has garnered significant interest for its potential use in treating various psychiatric and neurological disorders, including treatment-resistant depression, obsessive-compulsive disorders, addiction, and PTSD (Figure 2). Numerous preclinical and clinical studies have explored its effects [82,83,84] (Table 2). The hallucinogenic and neurological effects of psilocybin are linked to its modulation of serotonergic receptors, particularly 5-HT1A, 5-HT2A, and 5-HT2C. However, the therapeutic mechanism remains unclear. In the context of NPS in AD, psilocybin is being investigated in an open-label pilot clinical trial for depression in individuals with MCI or early AD (NCT04123314).

### 3.14. Antioxidants and Anti-Inflammatory Drugs

Oxidative stress and inflammation have long been studied as key therapeutic targets, and are recognized as pathophysiological features of neurodegenerative diseases like AD and neuropsychiatric disorders such as depression, psychosis, and aggression (Table 2). High levels of reactive oxygen species (ROS) are driven by stress response, neuroinflammation, imbalances in neurotransmitter function, and disruptions in synaptic plasticity. Many studies have demonstrated that oxidative stress is elevated in the brains of patients with AD and that the levels of antioxidants such as glutathione (GSH) are lower in the plasma of schizophrenia and patients with psychotic behavior [145,146]. Studies have found a link between depressive symptoms and lower intake of antioxidants, including selenium, and vitamins (A, C, E, B6, folate, and B12) [147]. Recently, findings indicate that in cases of excessive aggression coupled with depression, oxidative stress, and inflammation increase along with mitochondrial dysfunction, potentially leading to reduced synaptic function and worsening depressive symptoms [148,149,150]. Additionally, research has reported that cognitive decline in socially isolated mice was mitigated by the administration of the antioxidant N-acetyl-cysteine, while anxiety and aberrant socialization were exacerbated by social defeat stress. This increase in ROS in the brain may ultimately lead to greater social impairment and depressive behavior. These findings suggest that antioxidants like N-acetyl-cysteine could help alleviate anxiety, aggression, and social deficits caused by social defeat stress [62,151]. Thus, elevated ROS activity is implicated in neuropsychiatric symptoms, including anxiety, depression, and aggression, and ROS scavenging agents have the potential to restore normal behavior by promoting neurochemical homeostasis. Notably, several antioxidant and anti-inflammatory agents, such as DHA (NCT036138844), Flos gossypii flavonoids (NCT05269173), quercetine (NCT05422885), edravone (NCT05323812), and PrimeC (NCT06185543) [16] are currently in clinical trials for AD treatment in 2024. Although these compounds are primarily being tested as disease-modifying drugs, those with antioxidant properties may emerge as promising treatments for AD-related NPSs.

### 3.15. Social Deficits in AD

In addition to agitation/aggression, depression, anxiety, sleep dysregulation, psychosis, and apathy, AD significantly impacts social behavior in affected individuals [58,152,153,154,155,156]. Social deficits and dysregulation of social interactions are common NPS associated with AD (Table 2). These challenges arise from cognitive decline and changes in brain function during the disease’s pathological process, affecting the patient’s ability to engage in and navigate social situations. Caregivers also face enormous challenges in managing these behaviors. Patients with AD often exhibit impaired social cognition, including difficulty recognizing social cues, facial expressions, and body language, which are critical for understanding others’ emotions and intentions. It also results in problems with social judgment, leading to inappropriate or disinhibited behaviors in social settings. Due to language impairments, including trouble finding the right words, following conversations, understanding what others are saying, using pragmatic language, as well as repetitive questioning or conversation, patients with AD find social engagements and communication to be stressful and difficult. Due to the challenges faced during social interactions, individuals with AD may withdraw from social engagements, leading to isolation and exacerbating social difficulties. Patients with AD often struggle to regulate emotions and behaviors appropriately in social contexts, leading to outbursts, crying, or inappropriate laughter—similar to what is observed in other psychiatric disorders, including schizophrenia. Shifts in personality traits and other NPS in AD, such as increased irritability, suspicion, or apathy, further affect social functioning. A recent meta-analysis revealed facial expression recognition deficits in FTD and AD, with overall emotion recognition being most frequently impaired, followed by recognition of anger in FTD and fear in AD [157]. Additionally, patients with behavioral variant AD (bvAD) exhibit greater impairments in social cognition and divergent eye movement patterns compared to patients with typical AD, and the severity of social problems is comparable in many senses to behavioral variant FTD (bvFTD) [158]. Conversely, social isolation and loneliness are regarded as risk factors for AD and can adversely affect cognitive well-being and AD manifestation in both humans and animals [62,159,160,161,162]. A recent study estimates that a prevalence of as much as 4% can be reduced with the prevention of social isolation in later life [163].

Molecular and cellular mechanisms underlying the social deficits in AD are also under active investigation, as is identifying targets for more precise therapeutic intervention. In the Tg2576 mouse model of AD, social memory deficits are associated with a decreased presence of the extracellular matrix perineuronal net (PNN) around parvalbumin-positive interneurons (PV) and dysregulated long-term synaptic plasticity in the CA2 area of the hippocampus [164]. Furthermore, direct application of growth factor neuregulin-1 (NRG1) in the affected area sufficiently restored both PV/PNN levels and social memory performance of these mice, again exemplifying the importance of the regulation of neural activity in the brain regions involved in social behaviors [164]. A group of researchers focused on the serotonergic modulation to improve memory and social deficits in AD. In a study using Wistar rats, selective 5-HT6 receptor antagonist SLV was administered to rats reared in isolation, which showed positive effects on memory performance in an object recognition test (ORT). In prenatally valproic acid-exposed rat offspring, a commonly used rodent model mimicking ASD-like symptoms such as social deficits and repetitive behaviors, SLV mitigated the behavioral deficits. Additionally, SLV fully reversed MK-801-induced deficits in the ORT and scopolamine-induced memory deficits, although it did not show beneficial effects on the cognitive performance of 12-month-old Tg2576 mice [126]. In addition to the serotonergic modulators, administration of the dopamine D1 receptor positive allosteric modulator DETQ improves cognition and social interaction in aged mice, regardless of the treatment period. The effect may be related to the increased acetylcholine efflux in cortical and hippocampal regions [165].

Oxytocin, released from the hypothalamus, is one of the well-known “social hormones,” along with vasopressin (Figure 2). It plays an important role in other brain functions, such as stress regulation, appetite control, lactation, and associative learning. Based on the finding that the gene encoding for oxytocin, *oxt,* is differentially methylated in the brain and blood of patients with AD [127,128], it is speculated that oxytocin may modulate memory loss in AD [129]. Interestingly, low dose of long-term (42 days) intranasal oxytocin treatment mitigates spatial and working memory deficits in female APP/PS1 mice, suggesting the potential therapeutic effects of oxytocin against AD [129]. As expected, APP/PS1 mice showed decreased sociability in the social interaction test, but surprisingly enough, oxytocin further deteriorated social interaction in the experimental animals [129]. Further studies might be needed to underscore the mechanism of social deficits in AD and develop better therapeutic tools for the intervention of this specific domain of NPS in AD. On the contrary, a pilot randomized, double-blind, placebo-controlled crossover trial using intranasal oxytocin treatment revealed moderate–to–large effect size improvements in participant health outcomes and core social cognitive functions, as well as a reduction in caregiver burden. However, no positive effects were associated with participant outcomes related to social cognition itself [166].

Due to the seemingly similar nature of social interaction deficits in AD and ASD, as well as the developmental and degenerative nature of these disorders, some researchers have focused on the possible common etiology, pathological mechanisms, and association between the two, especially focusing on the role of APP, Aβ, and secretase [167,168,169,170,171,172] (Table 3). Studies using conditional APP/APLP1/APLP2 triple KO (cTKO) mice, which lack the APP family in excitatory forebrain neurons from embryonic day 11.5 onwards [168], have demonstrated agenesis of the corpus callosum and disrupted hippocampal lamination—phenotypes often observed in ASD model mice. These cTKO mice also exhibit dysregulated synaptic structure and function, impairments in learning and memory, repetitive behaviors (such as rearing and climbing), impaired social communication, and deficits in social interaction, which are core symptoms of ASD. These findings underscore the essential role of APP in regulating synaptic development, which is crucial for cognitive function and social behaviors [168]. Aberrations in the APP pathway, whether increased or decreased direction, have been associated with ASD-like phenotypes in experimental animals as well as in clinical samples, suggesting a common etiological mechanism for understanding both ASD and the social deficits observed in AD. For instance, mechanisms involving ERK receptor activation on the PI3K/Akt/mTOR/Rho GTPase pathway have been implicated [168,169,170,173,174].

Recently, it has been reported that CNTNAP2, one of the key proteins implicated in the manifestation of ASD, can be processed by γ-secretase. Similar to the situation where the amyloid-β precursor protein (APP) undergoes successive cleavage by β- and γ-secretases to generate the amyloid β protein (Aβ), the secreted APP (sAPP), and the APP intracellular domain (AICD) acting on nuclear transcription of target genes, a motif within the transmembrane domain of CNTNAP2 was postulated to be a target of γ-secretase because it is highly homologous to the γ-secretase cleavage site of APP. Like AICD and notch intracellular domain (NICD) production and activation, CNTNAP2 is cleaved by γ-secretase to produce the CNTNAP2 intracellular domain (CICD), which can translocate into the nucleus to modulate gene expression. Moreover, viral delivery of CICD to the medial prefrontal cortex (mPFC) in Cntnap2-deficient (Cntnap2^−/−^) mice normalized social deficits of the knockout (KO) mice, suggesting the essential role of CICD in the modulation of social behavior [171]. Interestingly, intermittent hypoxia for 14 days in reduced oxygen concentration (8%) enhanced sociability and working memory in C57BL/6J mice, which has been associated with the up-regulation of CNTNAP2 expression [172].

Overall, these results suggest a common etiological mechanism of social deficits observed in both AD and ASD as well as the feasibility of identifying and validating new targets for more precise control of symptoms in both disorders. The input obtained from either field of neurological disorders into another may facilitate breakthrough findings in regulating these seemingly formidable neuropsychiatric symptoms.

## 4. Implication for Potential Future Research and Drug Development

Neuropsychiatric symptoms (NPS) frequently co-occur with other symptoms, often overlapping into different symptom clusters, which complicates the identification of distinct syndromes. Clarifying these syndromes is essential, as understanding the phenotype of NPS can lead to the identification of specific brain regions and neural circuits involved, offering insights into their neuropathogenesis. It remains unclear whether the prevalence of certain NPS is influenced more by genetic factors, medical comorbidities, lifestyle patterns, neurotransmitter system involvement, or brain atrophy and disconnection, such as in the prefrontal cortex. Further translational research based on these neurobiological changes will facilitate the development of more targeted drug treatments and non-pharmacological management techniques, thereby improving both patient outcomes and caregivers’ well-being.

### 4.1. Impact of NPS on Caregivers and Strategies for Support

NPS in AD significantly increases the emotional, physical, and social burdens on caregivers, often leading to burnout, depression, and social isolation. To effectively support the well-being of caregivers, a multi-faceted approach is essential that encompasses emotional support, practical education, respite care, and systemic interventions [175]. Emotional support can include counseling and stress-reduction techniques, practical education for training on NPS management, and understanding AD progression. Respite care plays a crucial role in reducing burnout, and systemic interventions such as financial assistance and policy advocacy can further alleviate burdens. Providing caregivers with training in behavioral management, access to support networks, and the backing of policies that prioritize their health can not only improve their quality of life but also the quality of care they provide to patients with AD. Addressing caregiver distress through behavior interventions and modifying potential triggers in the environment can also lead to improvements in the symptoms experienced by patients with AD [176]. Ultimately, the well-being of caregivers is deeply intertwined with that of the patients they care for, making it vital to prioritize caregiver support in managing the complexities of AD.

### 4.2. Limitations of Current Treatments for NPS

Current treatments for NPS in AD are limited by side effects, lack of sustained efficacy, and the need for more targeted approaches. Antidepressants, for example, can cause side effects such as gastrointestinal issues, insomnia, and dizziness, and they may even worsen cognitive decline. Antipsychotics and benzodiazepines, used for agitation and anxiety, carry significant side effects, including increased sedation, cardiovascular issues, and a higher mortality rate in elderly patients with dementia. Furthermore, many conventional treatments fail to manage NPS effectively. SSRIs and antipsychotics have not consistently shown significant benefits over placebo in patients with AD. Similarly, cholinesterase inhibitors and stimulants (e.g., methylphenidate) used for apathy have demonstrated limited and inconsistent efficacy. A key issue with current treatments is that they are broad-acting, primarily targeting neurotransmitter systems like serotonin or dopamine, without addressing the specific neurobiological mechanisms underlying NPS in AD. NPS are not simply psychiatric symptoms superimposed on dementia; they are intertwined with the disease’s pathology. Therefore, there is a need for treatments that target AD-specific pathways, such as neuroinflammation and tau pathology, rather than relying on generalized psychiatric drugs. Additionally, polypharmacy in patients with AD increases the risk of adverse drug interactions, side effects, and reduced medication compliance. Therefore, more targeted and multimodal approaches that combine pharmacological and non-pharmacological interventions are essential.

### 4.3. Potential Targets for Social Function in AD

Addressing social functional impairment in AD is an emerging area of research, as social deficits significantly impact both the quality of life for patients and the burden on caregivers. Potential therapeutic targets for improving social cognition and interaction include pharmacological agents and non-pharmacological interventions. The social bonding hormone oxytocin, known for enhancing social behaviors, has been studied in autism spectrum disorders and schizophrenia and could be a promising therapeutic target in AD to address social processing impairments. Oxytocin may promote prosocial behaviors and reduce anxiety, potentially improving patients’ ability to socially interact. Other pharmacological targets include serotonin modulators, such as 5-HT2A and 5-HT6 receptor antagonists (e.g., pimavanserin, masupirdine), which show promise for enhancing social cognition. The endocannabinoid system, which regulates mood, anxiety, and social behavior is another potential therapeutic avenue. Cannabinoid receptor agonists like nabilone may modulate social interaction through anxiolytic and mood-enhancing effects. Modulating dopamine pathways, particularly via drugs that act on D2 and D3 receptors (e.g., brexpiprazole), may also alleviate social deficits in AD. Psychedelics such as psilocybin, which modulate serotonin receptors, have shown potential for improving social interaction and emotional processing. Social cognitive enhancers, such as the antioxidant N-acetyl cysteine (NAC) and cholinergic agent KarXT, could further enhance cognitive function and social interaction by modulating glutamatergic signaling. Additionally, cognitive-behavioral therapy (CBT) and social enrichment programs, as non-pharmacological approaches, could promote social skills alongside pharmacological treatments.

### 4.4. Potential Application for Biomarker Development

NPS can be classified based on central nervous system and peripheral biomarkers, as well as genetic polymorphisms. Advances in molecular imaging, particularly through the use of various ligands, allow for the visualization of neurotransmitter activity and the measurement of receptor occupancy. These techniques provide critical insights into disease prediction and drug responsiveness, especially for psychiatric disorders such as depression [177]. By analyzing interactions between molecular imaging data and receptor occupancy, researchers can gain valuable information about disease progression and treatment efficacy. Recent studies have introduced innovative non-invasive peripheral biomarkers, driven by machine learning, that hold promise for early diagnosis and prognosis of neurological conditions. For instance, voice biomarkers are being used for PD prediction and evaluating topological natural language processing (NLP) transformers on text message data [178,179]. Moving forward, measuring biomarkers, especially non-invasive biomarkers using machine learning techniques, in non-demented controls and patients with MCI/AD both with and without NPS will be useful for diagnosing and classifying NPS. Additionally, such biomarker analyses will help understand the impact of NPS on dementia risk and progression.

### 4.5. Non-Pharmacological Approaches for Managing NPS

The treatment of NPS in neurological disorders has evolved from the early use of penicillin for encephalitis to recent atypical antipsychotic drugs aimed at alleviating psychotic symptoms in patients with dementia. However, the effectiveness of these medications is often limited, showing little difference compared to a placebo. Research suggests that non-pharmacological treatments can show potential benefits for managing NPS in patients with dementia. Specifically, targeting environmental factors and underlying medical conditions can help alleviate behavioral symptoms.

The non-pharmacological approach is the first line of treatment for managing agitation in AD, as it addresses the underlying causes of agitation and promotes overall well-being without the risks associated with medications. Agitation in AD often stems from environmental factors, unmet emotional or physical needs, and communication difficulties rather than purely neurochemical imbalances. Addressing these root causes through techniques like environmental modifications, structured routines, and meaningful activities can reduce agitation effectively. Caregivers can be trained in methods such as redirection, validation therapy, and ensuring that basic needs—such as hunger, pain management, and comfort—are met. There is evidence that music and activity can help to manage behavioral symptoms such as agitation in individuals with dementia; exercise, and pleasant experiences can reduce depression; and also family caregiver training can improve these symptoms [176,180]. Additionally, creating a calm and consistent environment reduces overstimulation, which is a common trigger for agitation in patients with AD. Beyond these approaches, tools like cognitive-behavioral therapy (CBT) and neuromodulation techniques such as transcranial magnetic stimulation (TMS) or transcranial direct current stimulation (tDCS) offer further non-pharmacological interventions, although much should be done to prove the effectiveness and safety of these neuromodulatory means.

When medications are necessary, they should be used in conjunction with these non-pharmacological interventions to optimize effectiveness and minimize the risks of drug therapies. Combining behavioral strategies, environmental adjustments, CBT, and neuromodulation techniques can lead to more sustainable symptom management, potentially allowing for lower medication doses and improved patient outcomes. This integrative approach not only improves overall well-being but also targets the root causes of agitation rather than solely relying on pharmacological solutions. Therefore, while medications play a role in managing behavioral symptoms such as agitation in AD, exploring non-pharmacological approaches first and using them alongside medications can help address the underlying causes of NPS and improve the patient’s overall well-being. However, the relationship between non-pharmacological treatments and pathology has not yet been clearly established.

### 4.6. Potential Benefits of Combining Therapeutic Approaches

Combining different therapeutic approaches for managing NPS in AD offers significant potential for improving patient outcomes, as no single treatment currently addresses the wide range of emotional, behavioral, and cognitive challenges associated with NPS. A multimodal strategy that integrates both pharmacological and non-pharmacological treatments may provide more comprehensive symptom management, particularly for issues such as agitation, aggression, depression, anxiety, apathy, and psychosis. The primary benefit of combining these approaches is the synergistic effects that address both the biological and psychosocial aspects of NPS in AD. Pharmacological treatments target neurochemical imbalances, while non-pharmacological interventions, such as cognitive-behavioral therapy and environmental modifications, help patients develop coping strategies, reduce distress, and improve their overall quality of life. This combined strategy allows for the simultaneous targeting of multiple symptoms, potentially reducing their severity and enhancing patients’ daily functioning. However, one major concern with pharmacological treatments is the risk of polypharmacy, particularly in elderly patients with AD. Multiple medications increase the risk of adverse drug interactions, side effects, and medication noncompliance. Therefore, careful medication reviews and regular monitoring are essential to mitigate these risks. While pharmacological treatments can provide relief for NPS, they are often insufficient when used alone and can be accompanied by significant adverse effects, emphasizing the importance of personalized treatment plans. These plans should balance the use of medications and behavioral interventions tailored to each patient’s specific symptom profile and disease stage. Non-pharmacological approaches for managing NPS in AD include environmental modification, such as adjusting lighting, noise levels, and room layouts, to minimize triggering agitation and confusion. Caregiver education and support programs are also integral to treatment plans, equipping caregivers with tools to manage symptoms effectively. Although there is substantial research on both pharmacological and non-pharmacological treatments individually, studies on their combined use remain relatively limited. Further clinical trials and long-term studies are necessary to better understand how to integrate these therapies effectively, assess their safety, and determine their long-term efficacy in managing NPS in AD.

### 4.7. Long-Term Outcomes of Different Treatments

The long-term outcomes of treatments for NPS in AD vary significantly based on the type of treatment type and the disease stage. Current therapies focus on symptom management rather than altering the disease’s progression. Antipsychotics and antidepressants provide short-term effects but carry substantial risks with long-term use, and they do not impact the course of the disease. Similarly, medications like dextromethorphan/quinidine and cholinesterase inhibitors offer some symptomatic benefits without affecting disease progression. Non-pharmacological interventions, while often resource-intensive, provide sustained improvements in quality of life without adverse effects associated with medications. Meanwhile, the potential for disease modification remains a focus of ongoing research, particularly through the development of anti-inflammatory and neuroprotective agents. Clinical trials are actively investigating disease-modifying therapies, such as monoclonal antibodies targeting amyloid-beta or tau proteins. Although these drugs do not directly target NPS, they may reduce the overall burden of AD pathology, potentially leading to a reduction in NPS severity over time. It is anticipated that combining currently available treatments with new biologics could enhance their disease-modifying effects. However, more evidence is required to determine whether these approaches can meaningfully alter the course of AD, particularly in relation to NPS.

### 4.8. Potential Therapeutic Research Directions

It is important to note that the etiology and biological characteristics of NPS in the general population may differ from those in AD. Furthermore, our current understanding of how AD and NPS exacerbate each other is still incomplete, although significant interactions between the two are acknowledged. To address these knowledge gaps, further research is needed to develop effective pharmacological treatments for NPS in patients with AD. Such interventions aim to alleviate economic burdens and minimize overall side effects in response to the increasing patient population. Future studies should evaluate more psychosocial interventions combined with mild antipsychotic medications to mitigate, halt, and ultimately prevent NPS. Public health experts and clinicians should work towards establishing standardized definitions for both the overall diagnosis and treatment of NPS in AD, as well as for each individual symptom, in order to reduce diagnostic and therapeutic delays.

Future research on NPS in AD could focus on several promising directions to enhance diagnosis, treatment, and patient outcomes. First, identifying reliable biomarkers for NPS in AD is essential for improving early diagnosis and personalizing treatment. Potential biomarkers include neuroimaging techniques (e.g., PET, fMRI) to detect structural and functional changes, molecular biomarkers from blood or cerebrospinal fluid (CSF) markers, such as inflammatory markers, tau protein, and amyloid beta, as well as genetic markers like serotonin and dopamine polymorphisms, especially in relation to the manifestation and prognosis of NPS in AD. Second, research should focus on personalized treatment approaches that incorporate genetic, molecular, and clinical profiles. This includes the use of pharmacogenetics, patient subtyping, and lifestyle interventions to tailor treatments to individual needs. Third, integrating pharmacological treatments with non-pharmacological strategies, such as cognitive-behavioral therapy, environmental modifications, and social engagement, should be further explored to create comprehensive multimodal care plans for NPS. Lastly, long-term studies that track the progression of NPS alongside cognitive decline in AD can provide insights into symptom development and the impact of early intervention on disease progression. Advancing these research areas could lead to more predictive, precise, and individualized clinical approaches to NPS in AD, ultimately improving patients’ outcomes and quality of life as well as the development of more selective and safe therapeutics for managing NPS in AD.

## 5. Conclusions and Future Perspectives

This review provides several key takeaways for clinicians and patients that can improve the diagnosis, management, and treatment of NPS in AD. First, it suggests the importance of early diagnosis and screening of NPS in individuals at high risk of AD or showing MCI. NPS like depression, anxiety, and apathy often precede cognitive decline in AD. Clinicians should integrate routine screening tools for NPS in at-risk individuals, enabling earlier interventions that may help slow disease progression. Second, this review urges the appreciation of the limitations of current treatments. Conventional therapies, such as antidepressants and antipsychotics, often fail to effectively treat NPS in AD. Newer drugs like dextromethorphan/quinidine and brexpiprazole offer alternative options that target broader neurochemical pathways. Clinicians should consider these options, especially when traditional medications are ineffective. Continued interest and vigilance in this field of medicine is crucial. Third, this review provides the value of non-pharmacological approaches to treating NPS in AD. Behavioral interventions, such as cognitive-behavioral therapy for insomnia (CBT-I) and social support programs, can be valuable, especially for managing sleep disturbances, anxiety, and social deficits. Training caregivers in behavioral strategies such as recognizing early signs of agitation or aggression and using de-escalation techniques can alleviate both caregiver burden and patient distress. Integrating these approaches into care plans can reduce reliance on medication and improve quality of life. Fourth, the importance of social engagement in AD management is emphasized. Social isolation can accelerate disease progression, while increased social interaction, through structured activities or caregiver-supported engagement, can improve both cognitive and emotional outcomes. Clinicians should encourage maintaining or enhancing social interactions and explore interventions, both pharmacological and non-pharmacological, to enhance social functioning in patients with AD. Lastly, the review highlights the ongoing exploration of emerging therapies. New treatments, including psilocybin, pimavanserin, and AVP-786, are being investigated in clinical trials, with the potential for future expansion. These developments may offer new options for patients and their caregivers, fostering informed decision-making. In summary, clinicians can use these insights to better identify and treat NPS in AD, exploring newer pharmacological treatments, prioritizing non-pharmacological strategies, and focusing on social engagement to improve patient outcomes.

Alzheimer’s disease (AD), along with cognitive decline, is a major type of dementia, showing high prevalence rates worldwide due to the aging population and societal progress, with patients increasing over time. Numerous studies have been conducted to diagnose and treat AD, focusing on target discovery and drug development. Not only global big pharma but also startup venture companies worldwide are actively developing pipelines for AD. Currently, AD treatment pipelines are being developed based on disease-modifying drugs, cognitive-enhancing drugs, and neuropsychiatric symptom treatment.

Traditionally, AD therapeutic development has focused on pathology-based targets, aiming to address cognitive impairment and prevent neuronal cell death. Acetylcholinesterase inhibitors like donepezil, galantamine, and rivastigmine are currently used for cognitive impairment in AD. Recently, the US FDA approved AD therapies targeting amyloid beta clearance, including aducanumab in 2021, lecanemab in 2023, and donanemab in 2024. Additionally, clinical trials are underway for tau pathology and neuronal cell death inhibition.

Beyond cognitive impairment, neuropsychiatric symptoms such as agitation, aggression, apathy, psychosis, sleep disturbances, and social deficits significantly impact patients with AD and caregivers. In this review, the pipelines for NPS treatment were discussed, focusing on drugs targeting neurotransmitter receptors such as NMDA receptors, cannabinoids, sigma receptors, and adrenergic receptors. These drugs include cannabidiol, THC-free CBD, masupirdine, dronabidol, hallucinogens like psilocybin, and antioxidant molecules such as DHA, which are currently undergoing clinical trials.

In particular, symptoms such as agitation, anxiety, aggression, and social deficits observed in patients with AD are similar to those seen in Autism Spectrum Disorder (ASD). As a result, drugs that were previously tested as ASD treatments are now being investigated in clinical trials as potential NPS therapies for AD.

This potential trend extends from ASD, which exhibits social processing dysfunction based on the Research Domain Criteria (RDoC), to other neurological disorders such as depression and Parkinson’s disease.

Successful NPS treatment of AD is not only applicable for the sophisticated management of AD but may also have potential applications and help mechanistic understanding of other neurological disorders, such as depression, bipolar disorders, anxiety, schizophrenia, ASD, and Parkinson’s disease.

## Figures and Tables

**Figure 1 ijms-25-11338-f001:**
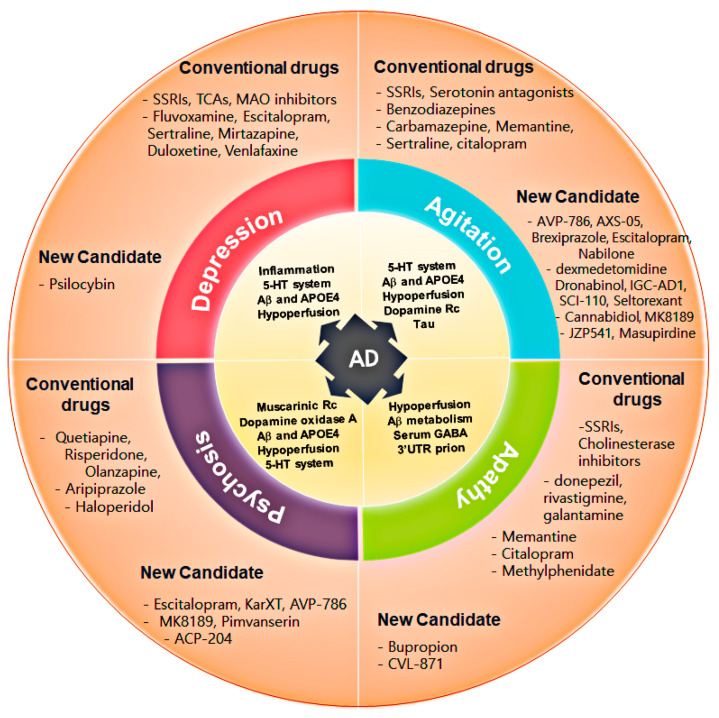
Behavioral and Psychological Symptoms in Alzheimer’s Disease [15,16,17].

**Figure 2 ijms-25-11338-f002:**
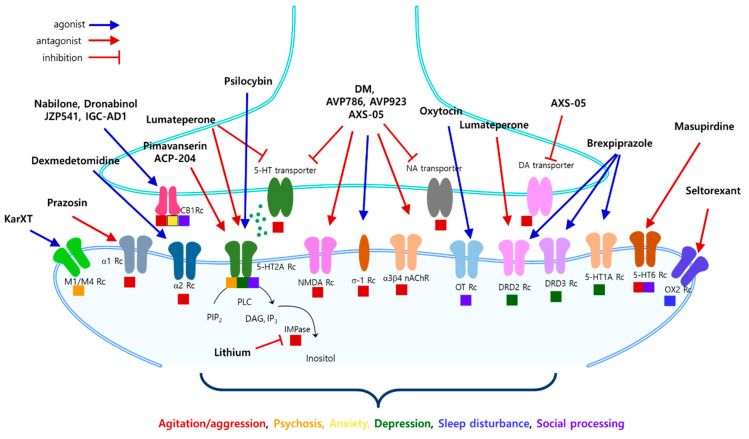
Summary of mechanism of candidate drugs for NPS treatment of AD. M1/M4 Rc; M1M4 muscarinic acetylcholine receptor, α1 Rc; alpha-1 adrenergic receptor, α2 Rc; alpha-2 adrenergic receptor, σ1 Rc; sigma 1 receptor, α3β4 nAChR; α3β4 nicotinic acetylcholine receptor, NA Rc; noradrenergic receptor, DRD2 Rc; Dopamine receptor D2, DRD3 Rc; Dopamine receptor D3, 5-HT1A Rc; serotonin 1A receptor, 5-HT2A Rc; serotonin 2A rèceptor, 5-HT6 Rc; serotonin 6 receptor, OX2 Rc; orexin 2 receptor, DM; Dextromethorphan. Each color represents the targets for each candidate drug.

**Table 1 ijms-25-11338-t001:** Comparison of each neuropsychiatric symptom of patients with AD and general patients.

NPS	Category	General Patients	Patients with AD
Depression	Prevalence	5–8.3%	14.8–40%
	Symptoms	Feeling sad, irritable, empty, disrupted sleep, hopelessness about the future, poor concentration, weight change, low energy, excessive guilt, or low self-worth	Persistent feelings of worthlessness, excessive/inappropriate guilt, a pervasive loss of interest or pleasure in all activities
	Treatment	SSRIs (citalopram, fluoxetine, paroxetine, sertraline), SNRIs (duloxetine, vanlafaxine, levomilnacipran), Atypical antidepressants (bupropion, mirtazapine, trazodone, vortioxetine), TCAs (imipramine, desipramine, amitriptyline), MAOIs (selegiline)	SSRIs (fluoxetine, paroxetine, sertraline, citalopram, escitalopram)
	Side effects	Nausea and vomiting, agitation, anxiety, indigestion, diarrhea or constipation, loss of appetite and weight loss, dizziness, blurred vision, dry mouth, excessive sweating, insomnia or drowsiness, headaches, and sexual side effects	No significant effects
Anxiety	Prevalence	4%	39%
	Symptoms	Trouble concentrating or making decisions, feeling irritable, tense or restless, nausea or abdominal distress, heart palpitations, sweating, trembling or shaking, and sleep problem	Excessive worry, fear, restlessness, and irritability
	Treatment	Benzodiazepines (alprazolam, diazepam, lorazepam), SSRIs, buspirone, TCAs (imipramine, clomipramine), MAOIs	SSRIs, SNRIs (venlafaxine), serotonergic atypical anxiolytics (buspirone), citalopram
	Side effects	Problems with balance and memory, drowsiness, confusion, vision problems, headaches, feelings of depression, dizziness, nausea, constipation, urinary retention, constipation	Abnormal bleeding, seizures, headaches, nausea, sleep trouble
Apathy	Prevalence	26–82%	49%
	Symptoms	Feeling flat, blunt, or numb emotionally, lack of emotional reaction, low energy and motivation, lack of goal setting, less interest in pleasure activities, hobbies, and relationships, anhedonia, lethargy	A decline in motivation and interest across emotional, goal-directed behavior, and cognitive activity
	Treatment	Antidepressants (trazodone, deprenyl, fluvoxamine), psychostimulants (methylphenidate, amphetamine), antipsychotics (risperidone), acetylcholinesterase inhibitors (donepezil, rivastigmine), NMDA receptor antagonists (memantine)	Cholinesterase inhibitors (donepezil, galantamine, rivastigmine), methylphenidate, Ginkgo biloba, modafinil, and SSRIs
	Side effects	Modest weight loss, no change in depression score	Weight loss and increased anxiety, no improvement, high blood pressure, cough, and osteoarticular pain
Agitation and Aggression	Prevalence	1–30%	30–60%
	Symptoms	Road rage, child abuse, sexual abuse, and domestic violence, verbal (swearing, shouting, or threatening), physical (hitting, punching, scratching, or biting)	Emotional distress, excessive psychomotor activity, aggressive behaviors, disruptive irritability, and disinhibition
	Treatment	Haloperidol, aripiprazole, droperidol, olanzapine,	Valproic acid, brexpiprazole, carbamazepine, SSRIs
	Side effects	Dizziness and nausea, paradoxical excitation, constipation, dry mouth, problems sleeping	Hepatotoxicity, GI upset, thrombocytopenia, coagulopathies, metabolic disorders, worsening cognitive dysfunction, agranulocytosis, cardiac arrhythmias
Psychosis	Prevalence	1.5–3.5%	50%
	Symptoms	Delusions, hallucinations, disorganized thought and behavior, poverty of speech, lack of energy, anhedonia, psychomotor retardation, catatonia	Delusions, hallucinations, disorganized thought and speech
	Treatment	Antipsychotics (clozapine and olanzapine), Benzodiazepines	Aripiprazole, risperidone, quetiapine, brexpiprazole
	Side effects	Drowsiness, dizziness, dry mouth, blurred vision, tiredness, nausea, constipation, weight gain, trouble sleeping, or muscle or nervous system problems (anxiety, agitation, jitteriness, drooling, trouble swallowing, restlessness, shaking, or stiffness)	Stroke, myocardial infarction
Sleep disturbances	Prevalence	20–41.7%	45%
	Symptoms	Excessive daytime sleepiness, irregular breathing or increased movement during sleep, depression, weight gain, lack of concentration, daytime fatigue, irritability, anxiety	Insomnia, excessive daytime sleepiness, nighttime awakenings, and alterations in sleep–wake patterns
	Treatment	Melatonin, zolpidem, zaleplon, eszopiclone, ramelteon, suvorexant, lamborexant or doxepin	Melatonin, trazodone, suvorexant, lemborexant
	Side effects	Changes in appetite, constipation or diarrhea, dizziness, headache, daytime drowsiness, heartburn, stomach pain, burning or tingling in the hands, arms, feet or legs, mental impairment	Various results, unclear effects
Social processing	Prevalence	7–11%	30–50%,
	Symptoms	Difficulty using appropriate greetings, changing language and communication style, telling and understanding stories, engaging in conversation, repairing communication breakdowns, using appropriated verbal and nonverbal signals, interpreting the verbal and nonverbal signals of others, making inferences, and forming and maintaining close relationships	Language problems, personality changes, and irritability
	Treatment	Behavior interventions, social communication treatments (comic strip conversations), social communication intervention, online speech therapy, social skills strengthening activities	N-acetyl cysteine, BDNF, NGF

**Table 3 ijms-25-11338-t003:** Comparison of mechanisms, symptoms, and therapeutic candidates of autism spectrum disorder and Alzheimer’s disease for social deficits.

	Alzheimer’s Disease	Autism Spectrum Disorder
**Symptoms**	Cognitive decline Learning and memory impairments Attention and perception change Impaired social cognition Impaired social judgement Language impairments Difficulty regulating emotions and behavior Depression Sleep disturbances Mood change: agitation, aggression, self-injurious behavior, impulsivity, hyperactivity, anxiety Personality change: irritability, suspiciousness, or apathy Social Isolation	Impaired communication Impaired reciprocal social interaction Restricted, repetitive, and stereotyped patterns of behaviors or interest learning difficulties Difficulty regulating emotions and behavior Dysfunction of working memory, planning, inhibition Depression Inattention Aggression Aggression, self-injurious behavior, impulsivity, irritability, hyperactivity, anxiety Sleep disturbance Social isolation
**Mechanisms**	Increase of Aβ peptide production and tau protein phosphorylation Altered synaptic plasticity Myelination changes Decrease of the extracellular matrix perineuronal net (PNN) Impairment of long-term synaptic plasticity in the hippocampus Neuregulin-1 (NRG1) decrease Impairment of serotonergic neurons Different methylation of oxytocin gene PI3K/Akt/mTOR/Rho GTPase pathway ERK/MAPK	Brain growth: neuronal migration Synaptic dysfunction E/I imbalance: GABA, NMDA/AMPA Abnormal 5-HT and dopamine neurotransmission Altered circuit connectivity PSD integrity: SHANK3, glutamate transmission Metabolic pathway: tryptophan metabolism, NADH, lactate/pyruvate ROS Neuroinflammation PI3K/Akt/mTOR, ERK/MAPK, calcium signaling Neuropeptide: decrease of vasopressin, oxytocin level
**Therapeutic candidates**	Growth factor (Neuregulin-1) SLV (5-HT6 receptor antagonist) DETQ (dopamine D1 receptor modulator) Oxytocin (social hormone) Antioxidant (N-acetyl cysteine) Neurotrophic factor (BDNF, NGF) Cannabidiol, nabilone, dronabinol, THC-Free CBD, IGC-AD1, JZP541 (cannabinoid) Psilocybin (5-HT1A,2A, 2C) Masupirdine (5-HT6) AVP-786, AXS-05, dextromethorphan (NMDARc antagonist) Dexmedetomidine (α2 adrenergic agonist)	Oxytocin (social hormone) ML-004 (5HT1b agonist) Brexpiprazole, caripiprazine, risperidone, aripiprazole (dopamine) EM-113, Arbaclofen (NMDARc antagonist) GWP42006, CBDV, JZP541 (cannabinoid) LSD, Psilocybin (5-HT1A,2A,2C) AB-2004, SB-121(microbiome) tDCS, rTMS (electroceutical)
**Common symptoms for social deficits**	Impaired social cognition: difficulty recognizing social cues, facial expressions, and body language. Problem with social judgment: inappropriate or disinhibited behaviors. Language impairments. Struggle to regulate emotions and behaviors: outbursts, crying, or inappropriate laughter. Personality change: irritability, suspicion, or apathy.	

## Data Availability

Not applicable.

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
