# Peer review of "Future Therapeutic Strategies for Alzheimer’s Disease: Focus on Behavioral and Psychological Symptoms"

_ijms, 2024, doi:10.3390/ijms252111338_

Round 1
Reviewer 1 Report
Comments and Suggestions for Authors
The article provides an overview of the current state of knowledge regarding neuropsychiatric symptoms (NPS) in Alzheimer's disease (AD). It discusses the prevalence, impact, and underlying mechanisms of NPS, as well as the available treatment options. The article also highlights the importance of non-pharmacological interventions and the need for further research to develop more effective treatments for these debilitating symptoms. The comments and suggestions below can be used to improve the quality of the presentation and scientific rigor of the article.
1. The authors should clarify the criteria used to select the included studies. A more detailed description of the search strategy, databases used, and inclusion/exclusion criteria would enhance the reproducibility of the review.
2. The authors should discuss the practical implications of the findings for clinicians and patients. How can the information presented in the review be used to improve the diagnosis, management, and treatment of NPS in AD?
3. The article could explore potential future research directions, such as the development of biomarkers for NPS in AD or the investigation of personalized treatment approaches based on individual patient characteristics.
4. The article could provide a more precise definition of NPS, as the term can be used in different ways. Clarifying the specific symptoms included in the definition would help to improve the clarity of the discussion. The authors could also discuss the factors that influence the prevalence of NPS in AD, such as disease stage, age of onset, and genetic factors.
5. The authors could provide a more detailed discussion of the neurobiological mechanisms underlying NPS in AD, including the role of neurotransmitters, brain regions, and neuropathological changes.
6. The article could discuss the limitations of current treatments for NPS in AD, such as side effects, lack of efficacy, and the need for more targeted approaches. The authors could discuss potential therapeutic targets for addressing social functional impairment in AD, such as drugs that modulate social cognition or facilitate social interaction.
7. The article could discuss the impact of NPS on caregivers and explore strategies for supporting their well-being.
8. The article could provide more information about the clinical trials evaluating the efficacy of dextromethorphan for NPS in AD.
9. The article could explore the potential benefits of combining different therapeutic approaches for NPS in AD.
10. I suggest adding these two references: “Voice Biomarkers for Parkinson's Disease Prediction Using Machine Learning Models with Improved Feature Reduction Techniques” and “Establishing an optimal online phishing detection method: evaluating topological NLP transformers on text message data”. By including these references, the authors can strengthen the scientific foundation of their article and provide additional insights into the potential applications of machine learning in Alzheimer's disease research.
11. The article could discuss the long-term outcomes of different treatments for NPS in AD and explore the potential for disease modification.
12. While medications play a role in managing agitation in AD, non-pharmacological approaches should be explored first and used alongside medications whenever possible. These interventions aim to address the underlying causes of agitation and improve the patient's overall well-being.
Author Response
Response to Reviewer’s comments
First of all, we would like to thank you for handling our manuscript. We appreciate the important points raised by this comment. We believe this comment helped a lot to improve the quality of the manuscript. We added this point throughout the manuscript. We tried to answer the suggestions raised by reviewers as faithfully as possible. We believe that expert comments from the reviewers helped a lot to improve the quality of the present article.
Point 1: The authors should clarify the criteria used to select the included studies. A more detailed description of the search strategy, databases used, and inclusion/exclusion criteria would enhance the reproducibility of the review.
Response 1: This review paper is not a systematic review based on meta-analysis, but rather a literature review focusing on recent research and drug development trends in NPS in AD. As such, the primary source of the studies covered in this study is articles which appeared in PubMed.
Point 2: The authors should discuss the practical implications of the findings for clinicians and patients. How can the information presented in the review be used to improve the diagnosis, management, and treatment of NPS in AD?
Response 2: Thank you for your helpful comment. This article provides a comprehensive review of the NPS associated with AD, emphasizing the clinical and therapeutic challenges in managing these symptoms. As per your comment, we have added this point into a new paragraph titled “5. Conclusion and future perspectives”.
- This review provides several key takeaways for clinicians and patients that can improve diagnosis, management, and treatment of NPS in AD. First, it suggests the importance of early diagnosis and screening of NPS in individuals at high risk of AD or showing MCI. NPS like depression, anxiety, and apathy often precede cognitive decline in AD. Clinicians should integrate routine screening tools for NPS in at-risk individuals, enabling earlier interventions that may help slow disease progression. Second, this review urges the appreciation the limitations of current treatments. Conventional therapies, such as antidepressants and antipsychotics, often fail to effectively treat NPS in AD. Newer drugs like dextromethorphan/quinidine and brexpiprazole offer alternative options that target broader neurochemical pathways. Clinicians should consider these options, especially when traditional medications are ineffective. Continued interests and vigilance in this field of medicine is crucial. Third, this review provides the value of non-pharmacological approaches to treating NPS in AD. Behavioral interventions, such as cognitive-behavioral therapy for insomnia (CBT-I) and social support programs, can be valuable, especially for managing sleep disturbances, anxiety, and social deficits. Training caregivers in behavioral strategies, such as recognizing early signs of agitation or aggression and using de-escalation techniques, can alleviate both caregiver burden and patient distress. Integrating these approaches into care plans can reduce reliance on medication and improve quality of life. Fourth, the importance of social engagement in AD management is emphasized. Social isolation can accelerate disease progression, while increased social interaction, through structured activities or caregiver-supported engagement, can improve both cognitive and emotional outcomes. Clinicians should encourage maintaining or enhancing social interactions and explore interventions, both pharmacological and non-pharmacological, to enhance social functioning in AD patients. Lastly, the review highlights the ongoing exploration of emerging therapies. New treatments, including psilocybin, pimavanserin, and AVP-786, are being investigated in clinical trials, with the potential for future expansion. These developments may offer new options for patients and their caregivers, fostering informed decision-making. In summary, clinicians can use these insights to better identify and treat NPS in AD, exploring newer pharmacological treatments, prioritizing non-pharmacological strategies, and focusing on social engagement to improve patient outcomes.
Point 3: The article could explore potential future research directions, such as the development of biomarkers for NPS in AD or the investigation of personalized treatment approaches based on individual patient characteristics.
Response 3: We have added this point into a new paragraph titled “4. Implication for potential future research and drug development”.
- Future research on NPS in AD could focus on several promising directions to enhance diagnosis, treatment, and patient outcomes. First, identifying reliable biomarkers for NPS in AD is essential for improving early diagnosis and personalizing treatment. Potential biomarkers include neuroimaging techniques (e.g., PET, fMRI) to detect structural and functional changes, molecular biomarkers from blood or cerebrospinal fluid (CSF) markers, such as inflammatory markers, tau protein, and amyloid beta, as well as genetic markers like serotonin and dopamine polymorphisms, especially in relation to the manifestation and prognosis of NPS in AD. Second, research should focus on personalized treatment approaches that incorporate genetic, molecular, and clinical profiles. This includes the use of pharmacogenetics, patient subtyping, and lifestyle interventions to tailor treatments of individual needs. Third, integrating pharmacological treatments with non-pharmacological strategies, such as cognitive-behavioral therapy, environmental modifications, and social engagement, should be further explored to create comprehensive multimodal care plans for NPS. Lastly, long-term studies that track the progression of NPS alongside cognitive decline in AD can provide insights into symptom development and the impact of early intervention on disease progression. Advancing these research areas could lead to more predictive, precise, and individualized clinical approaches to NPS in AD, ultimately improving patients’ outcomes and quality of life as well as development of more selective and safe therapeutics managing NPS in AD.
Point 4: The article could provide a more precise definition of NPS, as the term can be used in different ways. Clarifying the specific symptoms included in the definition would help to improve the clarity of the discussion. The authors could also discuss the factors that influence the prevalence of NPS in AD, such as disease stage, age of onset, and genetic factors.
Response 4: Thank you for your helpful comment. As per your comment, we added this point in the introduction section, and you can find the various symptoms of NPS listed in Table 1.
- Neuropsychiatric symptoms (NPS) are fundamental features of Alzheimer’s disease and related dementias, with nearly all individuals diagnosed with AD developing NPS during the disease progression. The prevalence of NPS in AD can vary depending factors such as disease stage, age of onset, and genetic predispositions. While NPS typically become more pronounced in the later stage of AD, population-based studies have shown that these symptoms often manifest in the very early stages, including the prodromal stage, such as MCI [1-5]. Among NPS, depression and apathy are the most frequently observed in patients with MCI and early AD, often accompanied by verbal and physical agitation. As AD advances, symptoms such as delusions, hallucinations, and aggression become more common, while apathy remains the most persistent and prevalent NPS across all stages of the disease. In addition, disruptions in circadian sleep-wake cycles become more pronounced in AD patients compared to normal aging processes. Notably, patients with MCI who experience depression are at a significantly higher risk of progressing to AD. Conversely, individuals with mild behavioral impairment are more likely to develop dementia, even if their cognitive function is within normal range. Symptoms like agitation, apathy, anxiety, disinhibition, euphoria, and irritability may have stronger associations with the onset of MCI than depression. Therefore, further research is needed to investigate the difference in prevalence associated with these factors.
Point 5: The authors could provide a more detailed discussion of the neurobiological mechanisms underlying NPS in AD, including the role of neurotransmitters, brain regions, and neuropathological changes.
Response 5: We appreciate your valuable comments. The neurobiological mechanisms underlying AD and NPS encompass the production of reactive oxygen species (ROS), neuroinflammation, alterations in energy metabolism, neuronal cell death, and structural changes in the brain, all contributing to neurofunctional decline. Neurotransmitters that play a critical role in neuropsychiatric symptoms (NPS) include alterations in serotonin, dopamine, norepinephrine, and acetylcholine levels. Affected brain regions, such as the prefrontal cortex, amygdala, nucleus accumbens, and hippocampus, are associated with neuronal structural changes, loss of connectivity, and modifications in neural circuits and axonal architecture. However, this encompasses an extensive amount of information and is essentially a separate topic for a paper. Therefore, we are currently in the process of drafting a distinct manuscript focused on the detailed mechanisms associated with NPS in AD.
- We have added a simplified illustration of the mechanisms of the drugs we introduced, which is included as figure 2 in the manuscript.
Figure 2. Summary of mechanism of candidate drugs for NPS treatment of AD. M1/M4 Rc; M1M4 muscarinic acetylcholine receptor, a1 Rc; alpha-1 adrenergic receptor, a2 Rc; alpha-2 adrenergic receptor, s1 Rc; sigma 1 receptor, a3b4 nAChR; a3b4 nicotinic acetylcholine receptor, NA Rc; noradrenergic receptor, DRD2 Rc; Dopamine receptor D2, DRD3 Rc; Dopamine receptor D3, 5-HT1A Rc; serotonin 1A receptor, 5-HT2A Rc; serotonin 2A re`ceptor, 5-HT6 Rc; serotonin 6 receptor, OX2 Rc; orexin 2 receptor, DM; Dextromethorphan. Each color represents the targets for each candidate drug.
Point 6: The article could discuss the limitations of current treatments for NPS in AD, such as side effects, lack of efficacy, and the need for more targeted approaches. The authors could discuss potential therapeutic targets for addressing social functional impairment in AD, such as drugs that modulate social cognition or facilitate social interaction.
Response 6: we appreciate your valuable comment. As per your comment, we have added these points into a new paragraph titled “4. Implication for potential future research and drug development”.
- Current treatments for NPS in AD are limited by side effects, lack of sustained efficacy, and the need for more targeted approaches. Antidepressants, for example, can cause side effects such as gastrointestinal issues, insomnia, and dizziness, and they may even worsen cognitive decline. Antipsychotics and benzodiazepines, used for agitation and anxiety, carry significant side effects including increased sedation, cardiovascular issues and a higher mortality rate in elderly dementia patients. Furthermore, many conventional treatments fail to manage NPS effectively. SSRIs and antipsychotics have not consistently shown significant benefits over placebo in AD patients. Similarly, cholinesterase inhibitors and stimulants (e.g., methylphenidate) used for apathy have demonstrated limited and inconsistent efficacy. A key issue with current treatments is that they are broad-acting, primarily targeting neurotransmitter systems like serotonin or dopamine, without addressing the specific neurobiological mechanisms underlying NPS in AD. NPS are not simply psychiatric symptoms superimposed on dementia; they are intertwined with the disease’s pathology. Therefore, there is a need for treatments that target AD-specific pathways, such as neuroinflammation and tau pathology, rather than relying on generalized psychiatric drugs. Additionally, polypharmacy in AD patients increases the risk of adverse drug interactions, side effects, and reduced medication compliance. Therefore, more targeted and multimodal approaches that combine pharmacological and non-pharmacological interventions are essential.
- Addressing social functional impairment in AD is an emerging area of research, as social deficits significantly impact both the quality of life for patients and the burden on caregivers. Potential therapeutic targets for improving social cognition and interaction include pharmacological agents and non-pharmacological interventions. The social bonding hormone oxytocin, known for enhancing social behaviors, has been studied in autism spectrum disorders and schizophrenia, and could be a promising therapeutic target in AD to address social processing impairments. Oxytocin may promote prosocial behaviors and reduce anxiety, potentially improving patients' ability to social interactions. Other pharmacological targets include serotonin modulators, such as 5-HT2A and 5-HT6 receptor antagonists (e.g., pimavanserin, masupirdine), which show promise for enhancing social cognition. The endocannabinoid system, which regulates mood, anxiety, and social behavior, is another potential therapeutic avenue. Cannabinoid receptor agonists like nabilone may modulate social interaction through anxiolytic and mood-enhancing effects. Modulating dopamine pathways, particularly via drugs that act on D2 and D3 receptors (e.g., brexpiprazole), may also alleviate social deficits in AD. Psychedelics such as psilocybin, which modulate serotonin receptors, have shown potential for improving social interaction and emotional processing. Social cognitive enhancers, such as the antioxidant N-acetyl cysteine (NAC) and cholinergic agent KarXT, could further enhance cognitive function and social interaction by modulating glutamatergic signaling. Additionally, cognitive-behavioral therapy (CBT) and social enrichment programs, as non-pharmacological approaches, could promote social skills, alongside pharmacological treatments.
Point 7: The article could discuss the impact of NPS on caregivers and explore strategies for supporting their well-being.
Response 7: As per your comment, we have added the following content into a new paragraph titled “4. Implication for potential future research and drug development”.
- NPS in AD significantly increase the emotional, physical, and social burdens on caregivers, often leading to burnout, depression, and social isolation. To effectively support the well-being of caregivers, a multi-faceted approach is essential-one that encompasses emotional support, practical education, respite care, and systemic interventions [6]. Emotional support can include counseling and stress-reduction techniques, practical education for training on NPS management and understating AD progression. Respite care plays a crucial role in reducing burnout, and systemic interventions such as financial assistance and policy advocacy can further alleviate burdens. Providing caregivers with training in behavioral management, access to support networks, and the backing of policies that prioritize their health, can not only improve their quality of life but also the quality of care they provide to AD patients. Addressing caregiver distress through behaviors interventions and modifying potential triggers in the environment can also lead to improvements in the symptoms experienced by AD patients [7]. Ultimately, the well-being of caregivers is deeply intertwined with that of the patients they care for, making it vital to prioritize caregiver support in managing the complexities of AD.
Point 8: The article could provide more information about the clinical trials evaluating the efficacy of dextromethorphan for NPS in AD.
Response 8: We added this point in the 3.1. section.
- 3.1.1. Dextromethorphan
Several key clinical trials have evaluated the efficacy of dextromethorphan, often in combination with other compounds like quinidine and bupropion to enhance its efficacy, for treating NPS in AD. The combination with bupropion (AXS-05) demonstrated therapeutic efficacy for depression, while the combination with quinidine (AVP-786, AVP-923) aimed to assess its efficacy for agitation [8, 9].
- 3.1.2. AVP-923
AVP-923 is a combination of dextromethorphan and quinidine, used to increase the bioavailability of dextromethorphan by inhibiting its metabolism via the cytochrome P450 enzyme (CYP2D6). Quinidine allows dextromethorphan to achieve therapeutic levels in the brain, where it can modulate various neurotransmitter systems.
The drug was generally well-tolerated, although side effects such as falls, diarrhea, and urinary tract infections were reported.
- 3.1.4. AXS-05
AXS-05 was well-tolerated, and the combination showed promise for its ability to manage NPS with fewer side effects compared to traditional antipsychotics.
Preliminary results suggest that AXS-05 may reduce the likelihood of agitation relapse, indicating its potential for long-term management of NPS in AD.
Point 9: The article could explore the potential benefits of combining different therapeutic approaches for NPS in AD.
Response 9: Thank you for your helpful comment. Combining different therapeutic approaches to manage NPS in AD offers a promising strategy for improving patient outcomes by addressing both the biological and psychosocial aspects of these symptoms. Pharmacological treatments, such as dextromethorphan/quinidine, SSRIs, or antipsychotics, can be paired with non-pharmacological interventions like environmental modifications, cognitive-behavioral therapy, and caregiver support to achieve a more holistic approach to care. However, the risks of polypharmacy, practical challenges in implementing non-pharmacological strategies, and the limited research on combined approaches must be carefully navigated. Future research needs to optimize combination strategies that provide comprehensive symptom relief while minimizing side effects, caregiver burden, and patient distress.
- As per your comment, we have added the following content into a new paragraph titled “4. Implication for potential future research and drug development”.
Combining different therapeutic approaches for managing NPS in AD offers significant potential for improving patient outcomes, as no single treatment currently addresses the wide range of emotional, behavioral, and cognitive challenges associated with NPS. A multimodal strategy that integrate both pharmacological and non-pharmacological treatments may provide more comprehensive symptom management, particularly for issues such as agitation, aggression, depression, anxiety, apathy, and psychosis.
The primary benefit of combining these approaches is the synergistic effects that address both the biological and psychosocial aspects of NPS in AD. Pharmacological treatments target neurochemical imbalances, while non-pharmacological interventions, such as cognitive-behavioral therapy and environmental modifications, help patients develop coping strategies, reduce distress, and improve their overall quality of life. This combined strategy allows for the simultaneous targeting of multiple symptoms, potentially reducing their severity and enhancing patients’ daily functioning.
However, one major concerns with pharmacological treatments is the risk of polypharmacy, particularly in elderly AD patients. Multiple medications increase the risk of adverse drug interactions, side effects, and medication noncompliance. Therefore, careful medication reviews and regular monitoring are essential to mitigate these risks. While pharmacological treatments can provide relief for NPS, they are often insufficient when used alone and can be accompanied by significant adverse effects, emphasizing the importance of personalized treatment plans. These plans should balance the use of medications and behavioral interventions tailored to each patient’s specific symptom profile and disease stage.
Non-pharmacological approaches for managing NPS in AD include environmental modification, such as adjusting lighting, noise levels, and room layouts, to minimize triggering for agitation and confusion. Caregiver education and support programs are also integral to treatment plans, equipping caregivers with the tool to manage symptoms effectively. Although there is substantial research on both pharmacological or non-pharmacological treatments individually, studies on their combined use is remain relatively limited. Further clinical trials and long-term studies are necessary to better understand how to integrated theses therapies effectively, assess their safety, and determine their long-term efficacy in managing NPS in AD.
Point 10: I suggest adding these two references: “Voice Biomarkers for Parkinson's Disease Prediction Using Machine Learning Models with Improved Feature Reduction Techniques” and “Establishing an optimal online phishing detection method: evaluating topological NLP transformers on text message data”. By including these references, the authors can strengthen the scientific foundation of their article and provide additional insights into the potential applications of machine learning in Alzheimer's disease research.
Response 10: Thank you for your helpful comment. As per your comment, machine learning (ML) is becoming an essential tool in AD research, with the potential to enhance early diagnosis, predict disease progression, and identify new therapeutic targets. By analyzing large datasets, ML algorithms can uncover patterns and biomarkers that are often missed by human analysis, which is vital due to AD's complexity and variability.
ML models can process neuroimaging data, such as MRI and PET scans, to detect early brain changes associated with AD, potentially identifying patients before clinical symptoms arise. Additionally, ML is applied to genomic and proteomic data to reveal new genetic risk factors and protein alterations linked to AD pathology. Another important use of ML is in analyzing electronic health records (EHRs), where it can predict AD progression based on patient history, cognitive assessments, and lifestyle factors. Natural language processing (NLP), a subset of ML, can also analyze unstructured clinical notes to extract relevant information on symptoms and patient outcomes, facilitating personalized treatment plans. In drug discovery, ML models help screen large chemical libraries and identify potential compounds targeting specific AD-related pathways. Overall, the integration of ML into AD research offers significant potential for improving our understanding and treatment of the disease. Thus, just as ML models are applied in AD research for early diagnosis, predicting disease progression, and discovering new therapeutic targets, they can also be utilized for early diagnosis, predicting disease progression, and identifying new therapeutic targets in neuropsychiatric symptoms (NPS), as well as for drug development. Additionally, this approach would be valuable for developing biomarkers that can differentiate NPS from other psychiatric disorders.
- We have added the following content, including these references you recommended, into a new paragraph titled “4. Implication for potential future research and drug development”.
NPS can be classified based on central nervous system and peripheral biomarkers, as well as genetic polymorphisms. Advances in molecular imaging, particularly through the use of various ligands, allow for the visualization of neurotransmitters activity and the measurement of receptor occupancy. These techniques provide critical insights into disease prediction and drug responsiveness, especially for psychiatric disorders such as depression [10]. By analyzing interactions between molecular imaging data and receptor occupancy, researchers can gain valuable information about disease progression and treatment efficacy. Recent studies have introduced innovative non-invasive peripheral biomarkers, driven by machine learning, which hold promise for early diagnosis and prognosis of neurological conditions. For instance, voice biomarkers are being for PD prediction and evaluating topological natural language processing (NLP) transformers on text message data [11, 12]. Moving forward, measuring biomarkers, especially non-invasive biomarkers using machine learning techniques, in non-demented controls and patients with MCI/AD both with and without NPS will be useful for diagnosing and classifying NPS. Additionally, such biomarker analyses will help understand the impact of NPS on dementia risk and progression.
Point 11: The article could discuss the long-term outcomes of different treatments for NPS in AD and explore the potential for disease modification.
Response 11: We appreciate your helpful comment. We have added this point into a new paragraph titled “4. Implication for potential future research and drug development”.
- The long-term outcomes of treatments for NPS in AD vary significantly based on the type of treatment type and the disease stage. Current therapies focus on symptom management rather than altering the disease’s progression. Antipsychotics and antidepressants provide short-term effects but carry substantial risks with long-term use, and they do not impact the course of the disease. Similarly, medications like dextromethorphan/quinidine and cholinesterase inhibitors offer some symptomatic benefits without affecting disease progression. Non-pharmacological interventions, while often resource-intensive, provide sustained improvements in quality of life without adverse effects associated with medications. Meanwhile, the potential for disease modification remains a focus of ongoing research, particularly through the development of anti-inflammatory and neuroprotective agents. Clinical trials are actively investigating disease-modifying therapies, such as monoclonal antibodies targeting amyloid-beta or tau proteins. Although these drugs do not directly target NPS, they may reduce the overall burden of AD pathology, potentially leading to a reduction in NPS severity over time. It is anticipated that the combining currently available treatments with new biologics could enhance their disease-modifying effects. However, more evidence is required to determine whether these approaches can meaningfully alter the course of AD, particularly in relation to NPS.
Point 12: While medications play a role in managing agitation in AD, non-pharmacological approaches should be explored first and used alongside medications whenever possible. These interventions aim to address the underlying causes of agitation and improve the patient's overall well-being.
Response 12: Thanks for the nice opinion on agitation and nonpharmacological treatment, which we cannot agree more. We have added the following content into a new paragraph titled “4. Implication for potential future research and drug development”.
- The treatment of NPS in neurological disorders has evolved significantly, from the early use of penicillin for encephalitis to modern atypical antipsychotic drugs aimed at reducing psychotic symptoms in dementia patients. However, the efficacy of these medications is often limited, frequently showing only modest benefits compared to a placebo. Recent research highlights the potential of non-pharmacological intervention in managing NPS, particularly in dementia patients, by addressing environmental factors and underlying medical conditions that contribute to behavioral symptoms.
For managing agitation in AD, non-pharmacological approaches should be the first line of treatment. These strategies aim to address the root causes of agitation, such as environmental factors, unmet emotional or physical needs, and communication difficulties, rather than focusing solely on neurochemical imbalances. Techniques such as modifying the environmental modifications, establishing structured routines, and engaging patients in meaningful activities have shown effectiveness in reducing agitation. Caregivers can be trained in strategies like redirection, validation therapy, and ensuring that basic needs, such as hunger, pain management, and comfort, are met. Evidence suggest that music and engaging activities can help to manage agitation, while exercise and pleasant experiences may reduce depression. Additionally, family caregiver training has shown to improve these symptoms in dementia patients [7, 13]. Creating a calm and consistent environment can also help reduce overstimulation, which is a common trigger for agitation in AD. Beyond these approaches, tools like Cognitive Behavioral Therapy (CBT) and neuromodulation techniques, such as transcranial magnetic stimulation (TMS) or transcranial direct current stimulation (tDCS), offer further potential, although more research is needed to establish their safety and efficacy in this context. When medications are necessary, they should be used in conjunction with non-pharmacological interventions to enhance effectiveness and minimize the risks with pharmacotherapy. This integrative approach-combining behavioral strategies, environmental adjustments, CBT, and neuromodulation-can lead to more sustainable symptom management, potentially allowing for lower medication doses and improved patient outcomes. Such approach not only improves overall well-being but also addresses remain important for managing behavioral symptoms in AD, prioritizing non-pharmacological treatment and using them alongside medications offers a more comprehensive approach. However, the relationship between non-pharmacological interventions and the underlying pathology of NPS in AD is not yet fully understood, requiring further investigation.
- The following are newly added paragraphs.
- Implication for potential future research and drug development
Neuropsychiatric symptoms (NPS) frequently co-occur with other symptoms, often overlapping into different symptom clusters, which complicates the identification of distinct syndromes. Clarifying these syndromes is essential, as understanding the phenotype of NPS can lead to the identification of specific brain regions and neural circuits involved, offering insights into their neuropathogenesis. It remains unclear whether the prevalence of certain NPS is influenced more by genetic factors, medical comorbidities, lifestyle patterns, neurotransmitter systems involvement, or brain atrophy and disconnection, such as in the prefrontal cortex. Further translational research based on these neurobiological changes will facilitate the development of more targeted drug treatments and nonpharmacological management techniques, thereby improving both patient outcomes and caregivers’ well-being.
4.1. Impact of NPS on caregivers and strategies for supporting
NPS in AD significantly increase the emotional, physical, and social burdens on caregivers, often leading to burnout, depression, and social isolation. To effectively support the well-being of caregivers, a multi-faceted approach is essential-one that encompasses emotional support, practical education, respite care, and systemic interventions [6]. Emotional support can include counseling and stress-reduction techniques, practical education for training on NPS management and understating AD progression. Respite care plays a crucial role in reducing burnout, and systemic interventions such as financial assistance and policy advocacy can further alleviate burdens. Providing caregivers with training in behavioral management, access to support networks, and the backing of policies that prioritize their health, can not only improve their quality of life but also the quality of care they provide to AD patients. Addressing caregiver distress through behaviors interventions and modifying potential triggers in the environment can also lead to improvements in the symptoms experienced by AD patients [7]. Ultimately, the well-being of caregivers is deeply intertwined with that of the patients they care for, making it vital to prioritize caregiver support in managing the complexities of AD.
4.2. limitations of current treatments for NPS
Current treatments for NPS in AD are limited by side effects, lack of sustained efficacy, and the need for more targeted approaches. Antidepressants, for example, can cause side effects such as gastrointestinal issues, insomnia, and dizziness, and they may even worsen cognitive decline. Antipsychotics and benzodiazepines, used for agitation and anxiety, carry significant side effects including increased sedation, cardiovascular issues and a higher mortality rate in elderly dementia patients. Furthermore, many conventional treatments fail to manage NPS effectively. SSRIs and antipsychotics have not consistently shown significant benefits over placebo in AD patients. Similarly, cholinesterase inhibitors and stimulants (e.g., methylphenidate) used for apathy have demonstrated limited and inconsistent efficacy. A key issue with current treatments is that they are broad-acting, primarily targeting neurotransmitter systems like serotonin or dopamine, without addressing the specific neurobiological mechanisms underlying NPS in AD. NPS are not simply psychiatric symptoms superimposed on dementia; they are intertwined with the disease’s pathology. Therefore, there is a need for treatments that target AD-specific pathways, such as neuroinflammation and tau pathology, rather than relying on generalized psychiatric drugs. Additionally, polypharmacy in AD patients increases the risk of adverse drug interactions, side effects, and reduced medication compliance. Therefore, more targeted and multimodal approaches that combine pharmacological and non-pharmacological interventions are essential.
4.3. Potential targets for social function in AD
Addressing social functional impairment in AD is an emerging area of research, as social deficits significantly impact both the quality of life for patients and the burden on caregivers. Potential therapeutic targets for improving social cognition and interaction include pharmacological agents and non-pharmacological interventions. The social bonding hormone oxytocin, known for enhancing social behaviors, has been studied in autism spectrum disorders and schizophrenia, and could be a promising therapeutic target in AD to address social processing impairments. Oxytocin may promote prosocial behaviors and reduce anxiety, potentially improving patients' ability to social interactions. Other pharmacological targets include serotonin modulators, such as 5-HT2A and 5-HT6 receptor antagonists (e.g., pimavanserin, masupirdine), which show promise for enhancing social cognition. The endocannabinoid system, which regulates mood, anxiety, and social behavior, is another potential therapeutic avenue. Cannabinoid receptor agonists like nabilone may modulate social interaction through anxiolytic and mood-enhancing effects. Modulating dopamine pathways, particularly via drugs that act on D2 and D3 receptors (e.g., brexpiprazole), may also alleviate social deficits in AD. Psychedelics such as psilocybin, which modulate serotonin receptors, have shown potential for improving social interaction and emotional processing. Social cognitive enhancers, such as the antioxidant N-acetyl cysteine (NAC) and cholinergic agent KarXT, could further enhance cognitive function and social interaction by modulating glutamatergic signaling. Additionally, cognitive-behavioral therapy (CBT) and social enrichment programs, as non-pharmacological approaches, could promote social skills, alongside pharmacological treatments.
4.4. Potential application for biomarkers development
NPS can be classified based on central nervous system and peripheral biomarkers, as well as genetic polymorphisms. Advances in molecular imaging, particularly through the use of various ligands, allow for the visualization of neurotransmitters activity and the measurement of receptor occupancy. These techniques provide critical insights into disease prediction and drug responsiveness, especially for psychiatric disorders such as depression [10]. By analyzing interactions between molecular imaging data and receptor occupancy, researchers can gain valuable information about disease progression and treatment efficacy. Recent studies have introduced innovative non-invasive peripheral biomarkers, driven by machine learning, which hold promise for early diagnosis and prognosis of neurological conditions. For instance, voice biomarkers are being for PD prediction and evaluating topological natural language processing (NLP) transformers on text message data [11, 12]. Moving forward, measuring biomarkers, especially non-invasive biomarkers using machine learning techniques, in non-demented controls and patients with MCI/AD both with and without NPS will be useful for diagnosing and classifying NPS. Additionally, such biomarker analyses will help understand the impact of NPS on dementia risk and progression.
4.5. Non-pharmacological approaches for managing NPS
The treatment of NPS in neurological disorders has evolved from early use of penicillin for encephalitis to recent atypical antipsychotic drugs aimed at alleviating psychotic symptoms in dementia patients. However, the effectiveness of these medications is often limited, showing little difference compared to a placebo. Research suggests that non-pharmacological treatments can show potential benefits for managing NPS in dementia patients. Specifically, targeting environmental factors and underlying medical conditions can help alleviate behavioral symptoms.
Non-pharmacological approach is the first line of treatment for managing agitation in AD, as they address the underlying causes of agitation and promote overall well-being without the risks associated with medications. Agitation in AD often stems from environmental factors, unmet emotional or physical needs, and communication difficulties rather than purely neurochemical imbalances. Addressing these root causes through techniques like environmental modifications, structured routines, and meaningful activities can reduce agitation effectively. Caregivers can be trained in methods such as redirection, validation therapy, and ensuring that basic needs—such as hunger, pain management, and comfort—are met. There are evidences that music and activity can help to manage behavioral symptoms such as agitation in individuals with dementia, exercise and pleasant experiences can reduce depression, and also family caregiver training can improve these symptoms [7, 13]. Additionally, creating a calm and consistent environment reduces overstimulation, which is a common trigger for agitation in AD patients. Beyond these approaches, tools like Cognitive Behavioral Therapy (CBT) and neuromodulation techniques such as transcranial magnetic stimulation (TMS) or transcranial direct current stimulation (tDCS) offer further non-pharmacological interventions although much should be done to prove the effectiveness and safety of these neuromodulatory means.
When medications are necessary, they should be used in conjunction with these non-pharmacological interventions to optimize effectiveness and minimize the risks of drug therapies. Combining behavioral strategies, environmental adjustments, CBT, and neuromodulation techniques can lead to more sustainable symptom management, potentially allowing for lower medication doses and improved patient outcomes. This integrative approach not only improves overall well-being but also targets the root causes of agitation rather than solely relying on pharmacological solutions. Therefore, while medications play a role in managing behavioral symptoms such as agitation in AD, exploring non-pharmacological approaches first and using them alongside medications can help address the underlying causes of NPS and improve the patient's overall well-being. However, the relationship between non-pharmacological treatments and pathology has not yet been clearly established.
4.6. Potential benefits of combining therapeutic approaches
Combining different therapeutic approaches for managing NPS in AD offers significant potential for improving patient outcomes, as no single treatment currently addresses the wide range of emotional, behavioral, and cognitive challenges associated with NPS. A multimodal strategy that integrates both pharmacological and non-pharmacological treatments may provide more comprehensive symptom management, particularly for issues such as agitation, aggression, depression, anxiety, apathy, and psychosis. The primary benefit of combining these approaches is the synergistic effects that address both the biological and psychosocial aspects of NPS in AD. Pharmacological treatments target neurochemical imbalances, while non-pharmacological interventions, such as cognitive-behavioral therapy and environmental modifications, help patients develop coping strategies, reduce distress, and improve their overall quality of life. This combined strategy allows for the simultaneous targeting of multiple symptoms, potentially reducing their severity and enhancing patients’ daily functioning. However, one major concerns with pharmacological treatments is the risk of polypharmacy, particularly in elderly AD patients. Multiple medications increase the risk of adverse drug interactions, side effects, and medication noncompliance. Therefore, careful medication reviews and regular monitoring are essential to mitigate these risks. While pharmacological treatments can provide relief for NPS, they are often insufficient when used alone and can be accompanied by significant adverse effects, emphasizing the importance of personalized treatment plans. These plans should balance the use of medications and behavioral interventions tailored to each patient’s specific symptom profile and disease stage. Non-pharmacological approaches for managing NPS in AD include environmental modification, such as adjusting lighting, noise levels, and room layouts, to minimize triggering for agitation and confusion. Caregiver education and support programs are also integral to treatment plans, equipping caregivers with the tool to manage symptoms effectively. Although there is substantial research on both pharmacological or non-pharmacological treatments individually, studies on their combined use is remain relatively limited. Further clinical trials and long-term studies are necessary to better understand how to integrated theses therapies effectively, assess their safety, and determine their long-term efficacy in managing NPS in AD.
4.7. Long-term outcomes of different treatment
The long-term outcomes of treatments for NPS in AD vary significantly based on the type of treatment type and the disease stage. Current therapies focus on symptom management rather than altering the disease’s progression. Antipsychotics and antidepressants provide short-term effects but carry substantial risks with long-term use, and they do not impact the course of the disease. Similarly, medications like dextromethorphan/quinidine and cholinesterase inhibitors offer some symptomatic benefits without affecting disease progression. Non-pharmacological interventions, while often resource-intensive, provide sustained improvements in quality of life without adverse effects associated with medications. Meanwhile, the potential for disease modification remains a focus of ongoing research, particularly through the development of anti-inflammatory and neuroprotective agents. Clinical trials are actively investigating disease-modifying therapies, such as monoclonal antibodies targeting amyloid-beta or tau proteins. Although these drugs do not directly target NPS, they may reduce the overall burden of AD pathology, potentially leading to a reduction in NPS severity over time. It is anticipated that the combining currently available treatments with new biologics could enhance their disease-modifying effects. However, more evidence is required to determine whether these approaches can meaningfully alter the course of AD, particularly in relation to NPS.
4.8. Potential therapeutic research directions
It is important to note that the etiology and biological characteristics of NPS in the general population may differ from those in AD. Furthermore, our current understanding of how AD and NPS exacerbate each other is still incomplete, although significant interactions between the two are acknowledged. To address these knowledge gaps, further research is needed to develop effective pharmacological treatments for NPS in AD patients. Such interventions aim to alleviate economic burdens and minimize overall side effects in response to the increasing patient population. Future studies should evaluate more psychosocial interventions combined with mild antipsychotic medications to mitigate, halt, and ultimately prevent NPS. Public health experts and clinicians should work towards establishing standardized definitions for both the overall diagnosis and treatment of NPS in AD, as well as for each individual symptom, in order to reduce diagnostic and therapeutic delays.
Future research on NPS in AD could focus on several promising directions to enhance diagnosis, treatment, and patient outcomes. First, identifying reliable biomarkers for NPS in AD is essential for improving early diagnosis and personalizing treatment. Potential biomarkers include neuroimaging techniques (e.g., PET, fMRI) to detect structural and functional changes, molecular biomarkers from blood or cerebrospinal fluid (CSF) markers, such as inflammatory markers, tau protein, and amyloid beta, as well as genetic markers like serotonin and dopamine polymorphisms, especially in relation to the manifestation and prognosis of NPS in AD. Second, research should focus on personalized treatment approaches that incorporate genetic, molecular, and clinical profiles. This includes the use of pharmacogenetics, patient subtyping, and lifestyle interventions to tailor treatments of individual needs. Third, integrating pharmacological treatments with non-pharmacological strategies, such as cognitive-behavioral therapy, environmental modifications, and social engagement, should be further explored to create comprehensive multimodal care plans for NPS. Lastly, long-term studies that track the progression of NPS alongside cognitive decline in AD can provide insights into symptom development and the impact of early intervention on disease progression. Advancing these research areas could lead to more predictive, precise, and individualized clinical approaches to NPS in AD, ultimately improving patients’ outcomes and quality of life as well as development of more selective and safe therapeutics managing NPS in AD.
References
- Lyketsos, C.G., et al., Neuropsychiatric symptoms in Alzheimer's disease. Alzheimers Dement, 2011. 7(5): p. 532-9.
- Hwang, T.J., et al., Mild cognitive impairment is associated with characteristic neuropsychiatric symptoms. Alzheimer Dis Assoc Disord, 2004. 18(1): p. 17-21.
- Lyketsos, C.G., et al., Prevalence of neuropsychiatric symptoms in dementia and mild cognitive impairment: results from the cardiovascular health study. JAMA, 2002. 288(12): p. 1475-83.
- Duara, R. and W. Barker, Heterogeneity in Alzheimer's Disease Diagnosis and Progression Rates: Implications for Therapeutic Trials. Neurotherapeutics, 2022. 19(1): p. 8-25.
- Poulin, S.P., et al., Risk Factors, Neuroanatomical Correlates, and Outcome of Neuropsychiatric Symptoms in Alzheimer's Disease. J Alzheimers Dis, 2017. 60(2): p. 483-493.
- Allegri, R.F., et al., Neuropsychiatric symptoms as a predictor of caregiver burden in Alzheimer's disease. Neuropsychiatr Dis Treat, 2006. 2(1): p. 105-10.
- Gitlin, L.N., et al., Targeting and managing behavioral symptoms in individuals with dementia: a randomized trial of a nonpharmacological intervention. J Am Geriatr Soc, 2010. 58(8): p. 1465-74.
- Akbar, D., et al., Dextromethorphan-Bupropion for the Treatment of Depression: A Systematic Review of Efficacy and Safety in Clinical Trials. CNS Drugs, 2023. 37(10): p. 867-881.
- Garay, R.P. and G.T. Grossberg, AVP-786 for the treatment of agitation in dementia of the Alzheimer's type. Expert Opin Investig Drugs, 2017. 26(1): p. 121-132.
- Lyketsos, C.G., et al., Developing new treatments for Alzheimer's disease: the who, what, when, and how of biomarker-guided therapies. Int Psychogeriatr, 2008. 20(5): p. 871-89.
- Nalini Chintalapudi , V.R.D., Gopi Battineni , Ciro Rucco and Francesco Amenta, Voice Biomarkers for Parkinson's Disease Prediction Using Machine Learning Models with Improved Feature Reduction Techniques. Vol. 1. 2023. 7.
- Baron, H.M.a.M., Establishing an Optimal Online Phishing Detection Method: Evaluating Topological NLP Transformers on Text Message Data. . Vol. 2. 2023. 9.
- Livingston, G., et al., Systematic review of psychological approaches to the management of neuropsychiatric symptoms of dementia. Am J Psychiatry, 2005. 162(11): p. 1996-2021.

Reviewer 2 Report
Comments and Suggestions for Authors
The authors' approach to defining AD from the perspective of future therapeutics is not only impressive but also holds significant potential to influence the field. The two different aspects they have highlighted are of utmost importance in our field.
However, the manuscript, while written in a descriptive way, stands as a good monograph in the field. This review should include the mechanistic aspects of the drugs that authors pose or recommend in a descriptive way. It is important to enhance the quality of the manuscript, particularly by using a graphical approach for each major drug they describe. This will not only improve the understanding of the researchers but also show that their work is respected and taken seriously in the field.
None at this point.
Author Response
Response to Reviewer’s comments
First of all, we would like to thank you for handling our manuscript. We appreciate the important points raised by this comment. We believe this comment helped a lot to improve the quality of the manuscript.
We have added a simplified illustration of the mechanisms of the drugs we introduced, which is included as figure 2 in the manuscript. We believe that expert comments from the reviewer helped a lot to improve the quality of the present article.
- The neurobiological mechanisms underlying AD and NPS encompass the production of reactive oxygen species (ROS), neuroinflammation, alterations in energy metabolism, neuronal cell death, and structural changes in the brain, all contributing to neurofunctional decline. Neurotransmitters that play a critical role in neuropsychiatric symptoms (NPS) include alterations in serotonin, dopamine, norepinephrine, and acetylcholine levels. Affected brain regions, such as the prefrontal cortex, amygdala, nucleus accumbens, and hippocampus, are associated with neuronal structural changes, loss of connectivity, and modifications in neural circuits and axonal architecture. However, this encompasses an extensive amount of information and is essentially a separate topic for a paper. Therefore, we are currently in the process of drafting a distinct manuscript focused on the detailed mechanisms associated with NPS in AD.
Figure 2. Summary of mechanism of candidate drugs for NPS treatment of AD. M1/M4 Rc; M1M4 muscarinic acetylcholine receptor, a1 Rc; alpha-1 adrenergic receptor, a2 Rc; alpha-2 adrenergic receptor, s1 Rc; sigma 1 receptor, a3b4 nAChR; a3b4 nicotinic acetylcholine receptor, NA Rc; noradrenergic receptor, DRD2 Rc; Dopamine receptor D2, DRD3 Rc; Dopamine receptor D3, 5-HT1A Rc; serotonin 1A receptor, 5-HT2A Rc; serotonin 2A re`ceptor, 5-HT6 Rc; serotonin 6 receptor, OX2 Rc; orexin 2 receptor, DM; Dextromethorphan. Each color represents the targets for each candidate drug.

Round 2
Reviewer 1 Report
Comments and Suggestions for Authors
Accept as is.
Reviewer 2 Report
Comments and Suggestions for Authors
Thanks to the authors for taking time to make thorough revision.